# Iterative Structural Inference of Directed Graphs

**Aoran Wang**
University of Luxembourg
`aoran.wang@uni.lu`

**Jun Pang**
University of Luxembourg
`jun.pang@uni.lu`

## Abstract

In this paper, we propose a variational model, *Iterative Structural Inference of Directed Graphs* (iSIDG), to infer the existence of *directed interactions* from observational agents' features over a time period in a dynamical system. First, the iterative process in our model feeds the learned interactions back to encourage our model to eliminate indirect interactions and to emphasize directional representation during learning. Second, we show that extra regularization terms in the objective function for smoothness, connectiveness, and sparsity prompt our model to infer a more realistic structure and to further eliminate indirect interactions. We evaluate iSIDG on various datasets including biological networks, simulated fMRI data, and physical simulations to demonstrate that our model is able to precisely infer the existence of interactions, and is significantly superior to baseline models.

## 1 Introduction

A wide range of dynamical systems in the real world can be viewed as the composition of interacting agents, including physical systems [24, 13], multi-agent systems [5, 27] and biological systems [48, 35]. Knowing the interactions of dynamical systems can promote our understanding of the intrinsic mechanisms of these systems, and further facilitate accurate prediction and control of dynamical systems. However, in many cases, unlike the states of the agents, the interactions are usually not directly observable or measurable. Therefore, it is necessary to uncover the underlying interactions of dynamical systems based on the observational states of the agents. In dynamical systems, the states of an agent are affected by the interactions, and the states are usually recorded as a set of continuous variables, which make it difficult to uncover the interactions based on the similarity between the agents. It is also worth mentioning that most dynamical systems are characterized by *directed interactions*, such as citation networks [44], social ego networks [31], and gene regulatory networks [35]. We address the revealing of the existence of interactions in the dynamical systems as the problem of structural inference, and model the dynamical system as a directed graph, in which the nodes represent the agents of the system, and the directed edges denote the interactions between the agents. Thus, the structural inference problem is transformed into the inference of the asymmetric adjacency matrix of the graph based on observational nodes' features (dynamics).

While several models have been proposed to address this problem with unsupervised variational generative frameworks [22, 51, 30], most of them only deal with undirected interactions, and almost none of them explains why the variational method succeeds in inferring the structure. In this paper, based on the theory of *Information Bottleneck* (IB) [47, 1], which states the training of variational auto-encoder (VAE) follows the two-step procedure of deep learning: label-fitting and representation-compression, we explain the intrinsic mechanism of structural inference with VAEs, and propose a novel VAE based machine learning approach, namely **i**terative **S**tructural **I**nference of **D**irected **G**raphs (iSIDG), for iterative inference of the directed graph structure based on observational node features. Our iSIDG model utilizes a graph neural network (GNN) as encoder and infers adjacency matrix of the graph in its latent space.

36th Conference on Neural Information Processing Systems (NeurIPS 2022).

The key rationale of our iSIDG model is the iterative process for structural inference, which is designed based on the Variational Information Bottleneck [1]. The iterative process not only feeds the learned structure backward with direction information, but also creates tighter bounds and relaxation during the training process. The direction information contributes to a more diverse edge representations in the encoder and latent space. The iterative process connects several rounds of training a VAE into a loop, and joints the second phase: representation-compression with the label-fitting phase of the next round, which creates a tighter bound and a relaxation, to learn a more precise disentanglement between indirect connections and direct ones. Unlike previous works [22, 51, 30] that totally rely on the bottleneck structure of VAE to infer structure, iSIDG has extra terms to regularize the learned features in the latent space, such as sparsity and connectiveness, which carefully takes the properties of (asymmetric) adjacency matrix into account. Experimental results show that iSIDG outperforms or matches the state-of-the-art baselines on datasets with directed or undirected graphs. More importantly, iSIDG can distinguish indirect connections from direct ones more precisely.

## 2 Related Work

Neural relational inference (NRI) [22] is the first to utilize a VAE to address the problem of structural inference based on observational node features. NRI conducts message-passing operations on a fixed fully connected graph structure in its encoder and infers graph structure in its latent space. Based on NRI, Webb et al. [51] propose factorized neural relational inference (fNRI), which extends NRI to multi-interaction systems. However, fNRI relies on ground-truth graph structure to classify results for each type of interaction. To deal with more complex systems, Li et al. [29] and Chen et al. [9] take various prior knowledge into account, such as sparsity and the distribution of node degrees, to increase the accuracy of structural inference. Unfortunately, it is quite difficult to acquire prior knowledge in many problem settings. Alet et al. [2] propose a modular meta-learning-based framework that jointly infers the structure with higher data efficiency. However, the framework has to be trained on various datasets beforehand to reach its best performance. From the aspect of Granger-causality, Löwe et al. [30] propose amortized causality discovery (ACD) method, which infers a latent posterior graph from temporal conditional dependence. Despite its high accuracy, the ACD method suffers from indirect interactions. Compared with this line of work, iSIDG introduces an iterative structural inference process to create a tighter bound by updating the fully connected graph structure in the encoder, and encourages edge representations to be diverse with the integrated direction information.

In addition to the aforementioned models to infer the structure of static graphs, several frameworks can also infer the structure of dynamic graphs. Ivanovic and Pavone [15] present a graph-structured model which combines recurrent sequence modeling and variational deep generative modeling to infer dynamic structure and predict future trajectories. Li et al. [26] propose a generic trajectory prediction forecasting network to provide multi-modal trajectory hypotheses and auxiliary structural inference. The framework in [12] utilizes sequential latent variable models to infer and predict separate relation graphs for every single time step. Moreover, the generic generative neural system proposed in [27] predicts trajectories based on dynamic graph representation and scene context information. The frameworks that can infer dynamic graph structures are mostly application-focused, and the structural inference is auxiliary to the task of trajectory prediction.

Besides the models based on variational interface, we would like to mention a few works which deal with the problem of structural inference as well, but based on different methodologies. ARNI framework [6] infers the latent structure based on regression analysis and a careful choice of basis functions. Mutual information is also utilized to determine the existence of causal links and thus can infer the structure of dynamical systems [52, 40]. Some approaches fit a dynamics model and then produce a causal graph estimate of the model by using recurrent models [45, 19], or infer the structure by generating edges sequentially [17, 28] and others independently prune the generated edges from an over-complete graph [41]. Xue and Bogdan [54] reconstruct network structures under unknown adversarial interventions. Although these methods can provide insights about the correlations between adjacent nodes, such as mutual information and causal relation, they either require prior knowledge, or demand the comparison of features from all possible node-pairs, or are sensitive to indirect interactions in the graphs.

There exists another branch of research called Graph Structure Learning, which aims to jointly learn an optimized graph structure and corresponding graph representations for downstream tasks [60, 11, 16]. However, these methods adopt similarity-based methods to optimize the noisy graph structure, which

cannot be applied to dynamical systems. Furthermore, the main focus of these methods is the downstream task, not the reconstruction of the graph structure. Besides that, there are also a series of work to reconstruct the structure of directed acyclic graphs [59, 57, 38, 58]. However, these works have difficulties to work on graphs with cycles (and self-loops), which are commonly observed among dynamical systems. Last but not least, in the research field of biology, there are some work which can infer the structure of gene regulatory networks [25, 20, 7]. But these works either require prior assumption or cannot be extended to dynamical systems whose agents have multi-dimensional features. Unlike these works, iSIDG focuses on the structural inference of dynamical systems, and can deal with cycles in the system and high-dimensional node features.

## 3 Preliminaries

**Notations and problem formulation.** We view a dynamical system as a directed graph, where the agents of the system are represented as the nodes of the graph, and the directed interactions between the agents as the edges. We note the directed graph as $\mathcal{G} = (\mathcal{V}, \mathcal{E})$, in which $\mathcal{V}$ represents the set of $n$ nodes: $\{v_i, 1 \leq i \leq n\}$, and $\mathcal{E}$ represents the set of edges: $(v_i, v_j) \in \mathcal{E} \subseteq \mathcal{V} \times \mathcal{V}$. Based on the set of edges, we derive to an asymmetric adjacency matrix $\mathbf{A} \in \mathbb{R}^{n \times n}$, in which $\mathbf{a}_{ij} \in \{0, 1\}$ indicates the presence ($\mathbf{a}_{ij} = 1$) of the edge between $v_i$ and $v_j$ or not ($\mathbf{a}_{ij} = 0$). The edges in our work are assumed to be static and do not evolve with time. The feature recordings of the nodes over a time period are represented as $\mathcal{V} = \{V^t, 0 \leq t \leq T\}$, where $T$ is the total number of time steps , and $V^t$ is the set of features of all the $n$ nodes at time step $t$: $V^t = \{v_0^t, v_1^t, \ldots, v_n^t\}$. Given $M$ sets of feature recordings $\{\mathcal{V}_k, 0 \leq k \leq M\}$, the structural inference problem we consider in this paper is to reconstruct the asymmetric adjacency matrix $\mathbf{A}$ of the graph in an unsupervised way.

We state the problem of structural inference as searching for a combinatorial distribution to describe the existence of the edges between any of the node pairs in the graph. As shown in Figure 1, we utilize a GNN based VAE [21] as the cornerstone, which can be trained to approximate complicated and high-dimensional probability distributions from the sampled node features [37]. We design an iterative process based on the IB theory, and feed the direction information, which is computed from the learned structure in the past training procedures, back to the input side, to encourage a more diverse learned representations, and to create a tighter bound as well as a relaxation of the training.

**Graph neural networks.** Graph neural networks (GNNs) are powerful deep learning algorithms to learn the representations of nodes and graphs. Most modern GNNs are designed based on the neighbourhood aggregation strategy [14], where the representation of a node is updated by aggregating the representations of its neighbours. In brief, we denote $h_i^{(k)}$ as the representation of node $v_i$ at the $k$-th layer, and the mechanism of these GNNs can be described as:

$$h_i^{(k)} = \texttt{UPDATE}^{(k)}\left(h_i^{(k-1)}, \mathbf{m}_{\mathcal{N}(v_i)}^{(k-1)}\right), \text{ where } \mathbf{m}_{\mathcal{N}(v_i)}^{(k-1)} = \texttt{AGGREGATE}^{(k)}(\{v_j \in \mathcal{N}(v_i)\}), \quad (1)$$

where $\mathcal{N}(v_i)$ denotes the set of chosen nodes related to $v_i$ (e.g., its neighbours), UPDATE denotes the arbitrary function in the layer to update node representation, and AGGREGATE represents the aggregation function in the layer.

GCN [23] updates node representation based on one-hop neighbours and also its own representation in the previous layer, and calculates new representations with ReLU. GIN [53] aggregates the representations of adjacent nodes and node's own representation in the previous layer as a weighted summation, and utilizes multilayer perceptron to update node representations. GAT [49] introduces attention mechanism to implicitly specify different weights to adjacent nodes, and then updates representations upon the weighted summation. GNNs are efficient in learning rich representations for nodes and graphs. As a result, we utilize GNNs in the encoder to firstly learn representations of the nodes, and later on learn edge representations from the node representations in iSIDG.

**Information bottleneck.** The theory of *Information Bottleneck* (IB) provides an explanation and understanding of learning with deep neural networks [47, 46, 42]. In general, for the input data $X$ and its label $Y$, the IB aims to learn the minimal sufficient representation $Z = \arg\min_Z I(Z; X) - \mathfrak{u} \cdot I(Z; Y)$, where $\mathfrak{u}$ is the Lagrangian multiplier to balance sufficiency and minimality. The training epochs of standard deep learning can be divided into two phases: label-fitting and representation-compression. These two phases can be characterized by the mutual information between the input features and the learned representation of a layer $I(X; Z)$, as well as between learned representation and the label variables $I(Z; Y)$. In the label-fitting phase, both $I(X; Z)$ and $I(Z; Y)$ increase, but

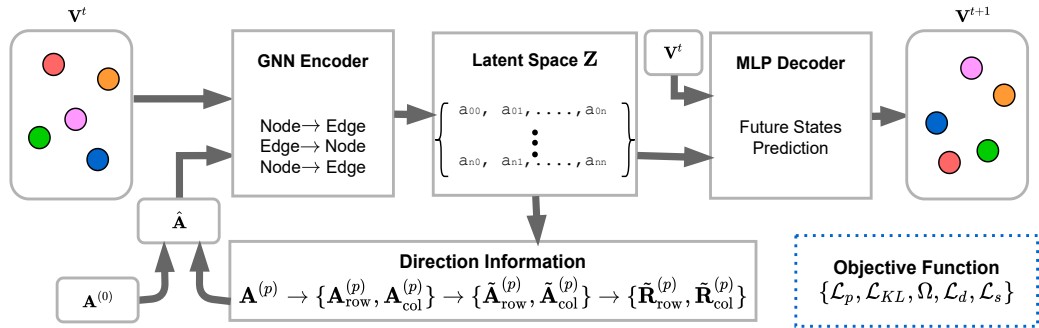

Figure 1: Overall architecture of the proposed iSIDG framework. iSIDG takes the fully-connected graph structure for the first run, but later on it takes the learned adjacency matrix with direction information as input. Regularization terms including smoothness, connectiveness and sparsity encourage learned features in latent space to be realistic and eliminate inferred indirect interactions.

in the representation-compression phase, $I(X; Z)$ decreases while $I(Z; Y)$ can either decrease or increase, depending on the amount of data [42].

Alemi et al. [1] present a variational approximation to the IB theory (VIB), and prove VAE objective is a special case of the variational approximation. Under a fully unsupervised setting, VAEs learn the minimal sufficient statistics from the input features, which are sufficient to derive the output features, and keep the most compressed and abstractive representations in their latent space. We utilize a VAE to infer the structure of the graph in its latent space, which is minimally sufficient to predict the future states of the nodes together with present node states.

It is suggested that IB iterations with stochastic relaxation methods may boost training [42]. It is also mentioned that a relaxation process is helpful to reduce local ambiguity and achieve global consistency [56], as already proved in previous work [8, 55]. Therefore, we design an iterative process with stochastic gradient descent (SGD) based optimizer to create a tighter bound and a relaxation for the training, and to encourage iSIDG to further disentangle dynamics from the adjacency matrix. In order to verify the effectiveness of this iterative process, we conduct a case study in Section D.3.

## 4 Model Design

### 4.1 Structural Inference by Information Bottleneck

Together with node features, graph structure characterizes the basic information of a graph, which represents a dynamical system. The goal of structural inference is to accurately reconstruct the adjacency matrix $\mathbf{A}$ of the directed graph based on the recordings of node features: $\mathbf{A} \leftarrow \{V^t, 0 \leq t \leq T\}$. Interestingly, for dynamical systems, the future features of the nodes are deterministic by their previous features and the connectivity: $V^{t+1} \leftarrow \{V^0, V^1, \ldots, V^t; \mathbf{A}\}$. We assume the Markovian assumption works in our case, and thus previous statement can be simplified to: $V^{t+1} \leftarrow \{V^t; \mathbf{A}\}$. Thus, we derive to an assumption: the information about the adjacency matrix is implicitly included in the node features over a time period, which also conforms to the IB theory.

Based on VIB, we derive the *Information Bottleneck for Structural Inference*, which aims at inferring the adjacency matrix $\mathbf{A}$ in the latent space $\mathbf{Z}$:

$$\mathbf{Z} = \arg \min_{\mathbf{Z}} I(\mathbf{Z}; V^t, \mathbf{A}) - \mathfrak{u} \cdot I(\mathbf{Z}; V^{t+1}), \tag{2}$$

Interestingly, $\mathbf{A}$ is also required in the first term in Equation 2, which is unrealistic in our problem setting. Thus, we assume a fully connected graphs structure $\mathbf{A}^{(0)}$ as input, which is not less informative than the actual sparse adjacency matrix:

$$\mathbf{Z} = \arg \min_{\mathbf{Z}} I(\mathbf{Z}; V^t, \mathbf{A}^{(0)}) - \mathfrak{u} \cdot I(\mathbf{Z}; V^{t+1}). \tag{3}$$

Following the statement in VIB [1], we adopt a VAE [21] to deal with the problem described in Equation 3, where an encoder $q_\phi(\mathbf{Z}|V^t, \mathbf{A}^{(0)})$ approximates $I(\mathbf{Z}; V^t, \mathbf{A}^{(0)})$, and a decoder

$p_\theta(\hat{V}^{t+1}|\mathbf{Z}, V^t)$ approximates $I(\mathbf{Z}; V^{t+1})$. $\phi$ and $\theta$ represent the parameters of encoder and decoder, respectively. We set the latent space of the VAE with the size of $n \times n$. The fixed dimension of the latent space restricts the size of the search space for the adjacency matrix. We utilize a weighted ELBO with multiple integrated regularization terms to optimize model parameters $\{\phi, \theta\}$.

## 4.2 Encoder

The encoder in our model is tasked with inferring the existence of directed interactions $\mathbf{z}_{ij}$ between all node-pairs by extracting the most compressed representation given the node features $\mathcal{V}$.

The encoder $q_\phi(\mathbf{Z}|V^t, \mathbf{A}^{(0)})$ returns a set $\{\mathbf{z}_{ij}|v_i, v_j \in \mathcal{V}\}$, which features the distribution to describe how probable the edge from node $j$ to node $i$ exists: $p(\mathbf{z}_{ij}) \in [0, 1]$. The prior $p(\mathbf{Z}) = \prod_{i,j \in \mathcal{V}} p(\mathbf{z}_{ij})$ is a factorized uniform distribution. We utilize GNNs to extract features in the encoder. But the GNNs learn new representation based on both node features and the topology of the graph $f_{\text{GNN}}(\mathcal{V}, \mathbf{A})$, in which the adjacency matrix $\mathbf{A}$ is our learning target. To encounter this problem, we impose an iterative learning process on the adjacency matrix, and we discuss the details in Section 4.4. For simplicity, in this section we use $\hat{\mathbf{A}}$ to represent the adjacency matrix. In general, the encoder:

$$q_\phi = \text{softmax}\left(\mathbf{f}_{\text{enc}}(\mathcal{V}, \hat{\mathbf{A}})\right), \tag{4}$$

where $f_{\text{enc}}$ represents the encoder. The basic setup of our encoder is similar to that in [22], which applies two rounds of node-to-edge and one round of edge-to-node message passing operations to learn the edge embeddings $\mathbf{h}_{ij}$:

$$\text{Node Embedding:} \quad \mathbf{e}_j^1 = f_{\text{embed}}^{(1)}(v_j), \tag{5}$$

$$\text{Node-to-edge:} \quad \mathbf{e}_{ij} = f_e^1([\mathbf{e}_i^1, \mathbf{e}_j^1]), \text{ if } a_{ij} = 1 \text{ in } \hat{\mathbf{A}}, \tag{6}$$

$$\text{Edge-to-node:} \quad \mathbf{e}_j^2 = f_v\left(\sum \mathbf{e}_{ij}\right), \tag{7}$$

$$\text{Node-to-edge:} \quad \mathbf{h}_{ij} = f_e^2([\mathbf{e}_i^2, \mathbf{e}_j^2]), \text{ if } a_{ij} = 1 \text{ in } \hat{\mathbf{A}}, \tag{8}$$

where $f_{\text{embed}}^{(1)}$ is the embedding network for input features, $f_e^1$ and $f_e^2$ are node-to-edge message-passing networks, and $f_v$ is the edge-to-node operation. We observe that Equations 5 - 7 is a round of message passing for node features in most GNNs, as a result, we substitute Equations 5 - 8 as:

$$\mathbf{h}_{ij} = f_e^2\left(f_{\text{GNN}}(\mathcal{V}, \hat{\mathbf{A}}), \hat{\mathbf{A}}\right), \tag{9}$$

so we can integrate state-of-the-art GNNs such as GCN, GAT and GIN into encoder. In this work, we integrate GIN into the encoder.

As a consequence, our encoder is designed to learn informative features from input node features and compress them to fit the restricted search space, which has the dimension of a fully connected graph. By extracting information from the high-dimensional but low-level node features to the restricted and low-dimensional latent space, our encoder disentangles the dynamics and approximates the minimally sufficient statistics, the adjacency matrix. After that, we model the existence of edges as $\mathbf{Z} \in \mathbb{R}^{n \times n}$, where its component $\mathbf{z}_{ij}$ comes from $q_\phi(\mathbf{z}_{ij}|V^t, \hat{\mathbf{A}}) = \text{softmax}(\mathbf{h}_{ij})$.

## 4.3 Decoder

The task of the decoder is to predict the future states $V^{t+1}$ based on the samplings from latent space $\mathbf{Z}$ and the present node features $V^t$. From the IB theory, if the encoder derives minimally sufficient statistics as the adjacency matrix, the decoder can precisely predict the future node features. However, there remains the problem of how to design the decoder, so that it can further encourage the encoder to disentangle dynamics and to infer the graph structure only. In order to deal with this problem, we design the decoder to imitate a single message-passing operation. We feed our decoder with the present node features besides the samplings from the latent space to predict the future node features. As a consequence, the node features are disentangled from the compressed information in the latent space during training. In short, the decoder deals with the following problem: $p_\theta(V^{t+1}|V^t, \mathbf{Z})$. We

use a two-stage decoder, which consists of a message-passing layer and an embedding layer:

$$\hat{\mathbf{h}}_j^{t+1} = v_j^t + f_{\text{embed}}^{(3)}\left( \sum_{\mathbf{z}_{ij}>0} \mathbf{z}_{ij} f_{\text{embed}}^{(2)}([v_i^t, v_j^t]) \right), \tag{10}$$

where $f_{\text{embed}}^{(2)}$ and $f_{\text{embed}}^{(3)}$ are multilayer perceptrons (MLPs) to embed the features.

## 4.4  Iterative Learning with Direction Information

The motivation of integrating iterative learning process into the structural inference consists of three aspects: (1) to create a tighter bound for Equation 3; (2) to create a relaxation for the training process; (3) to emphasize direction information. In order to formulate the ideas more precisely, we would like to firstly describe the iterative process with direction information in our model.

From the very beginning, we have no prior knowledge about the structure of the graph. In order to utilize GNNs to extract and compress the features into a restricted search space for adjacency matrix, we feed the encoder with a fully connected graph $\mathbf{A}^{(0)} = \mathbf{1} \in \mathbb{R}^{n \times n}$. After $\eta$ rounds of training, we obtain an asymmetric adjacency matrix $\mathbf{A}^{(p)}$ from the latent space, which is calculated from the averaged result of the entire training samples:

$$\mathbf{A}^{(p)} = \frac{1}{m} \sum_{k \in M} \mathbf{Z}_k, \tag{11}$$

where $m$ is the total number of training samples, and $M$ is the set of training data. The full-batch average operation not only preserves the most information from the training set, but also makes the $\mathbf{A}^{(p)}$ to be invulnerable to outliers. We then normalize $\mathbf{A}^{(t)}$ in both row-wise and column-wise:

$$\mathbf{A}_{\text{row}}^{(p)} = \mathbf{D}_{\text{in}}^{-1} \mathbf{A}^{(p)}, \text{ and } \mathbf{A}_{\text{col}}^{(p)} = \mathbf{A}^{(p)} \mathbf{D}_{\text{out}}^{-1}, \tag{12}$$

where $\mathbf{D}_{\text{in}}^{-1}$ and $\mathbf{D}_{\text{out}}^{-1}$ are in-degree and out-degree matrices which are calculated based on $\mathbf{A}^{(p)}$, respectively. We name Equations 12 as *direction information*. After that, we combine the direction information matrices with the normalized initial fully-connected adjacency matrix, respectively:

$$\tilde{\mathbf{A}}_{\text{row}}^{(p)} = \lambda \mathbf{L} + (1-\lambda)\mathbf{A}_{\text{row}}^{(p)}, \text{ and } \tilde{\mathbf{A}}_{\text{col}}^{(p)} = \lambda \mathbf{L} + (1-\lambda)\mathbf{A}_{\text{col}}^{(p)}, \tag{13}$$

where $\mathbf{L}$ is the normalized adjacency matrix of the initial fully-connected graph $\mathbf{A}^{(0)}$, and $\lambda$ is the combination coefficient. Since $\mathbf{A}^{(0)}$ is symmetric, we can use either column-wise or row-wise normalization. We map the results in Equation 13 to binary sets with the characteristic function:

$$\tilde{\mathbf{R}}_\kappa^{(p)} = \mathbb{1}_\xi(\tilde{\mathbf{A}}_\kappa^{(p)}) = \begin{cases} 1 & \text{if } \mathbf{a}_{ij} \geq \xi \text{ for } \mathbf{a}_{ij} \text{ in } \tilde{\mathbf{A}}_\kappa^{(p)} \\ 0 & \text{otherwise} \end{cases} \tag{14}$$

where $\kappa \in \{\text{row}, \text{col}\}$, and $\xi$ is the threshold value. Finally, we feed $\tilde{\mathbf{R}}_\kappa^{(p)}$ back to the input side of encoder mentioned in Section 4.2 by substituting the $\hat{\mathbf{A}}$ in Equations 6 and 8. More formally, the two feature vectors which are concatenated in Equations 6 and 8 represent the sender and receiver of the edge, respectively. We rewrite the equations for the iterative process according to $\tilde{\mathbf{R}}_\kappa^{(p)}$:

$$\mathbf{k}_{ij} = \begin{cases} f_e^n([\mathbf{k}_i, \mathbf{k}_j]) & \text{if } \mathbf{r}_{ij} = 1 \text{ in } \tilde{\mathbf{R}}_{\text{col}}^{(p)} \text{ and } \tilde{\mathbf{R}}_{\text{row}}^{(p)} \\ f_e^n([\mathbf{k}_i, 0]) & \text{if } \mathbf{r}_{ij} = 1 \text{ only in } \tilde{\mathbf{R}}_{\text{col}}^{(p)} \\ f_e^n([0, \mathbf{k}_j]) & \text{if } \mathbf{r}_{ij} = 1 \text{ only in } \tilde{\mathbf{R}}_{\text{row}}^{(p)}, \end{cases} \tag{15}$$

where $\mathbf{k}_{ij} \in \{\mathbf{e}_{ij}, \mathbf{h}_{ij}\}$, $f_e^n \in \{f_e^1, f_e^2\}$, and $\mathbf{k}_n \in \{\mathbf{e}_n^1, \mathbf{e}_n^2\}$ in Equations 6 and 8, respectively. We then train iSIDG with the updated adjacency matrices $\tilde{\mathbf{R}}_\kappa^{(p)}$. The stop condition for iteration is:

$$\|\mathbf{A}^{[i]} - \mathbf{A}^{[i-1]}\|_F^2 < \delta \|\mathbf{A}^{(0)}\|_F^2, \tag{16}$$

where superscript $[i]$ represents the rounds within current iteration. Once this condition is satisfied, the adjacency matrix at the input side of the encoder will be updated according to Equations 11 - 15.

**Tighter bound.** During training, the initial fully connected adjacency matrix $\mathbf{A}^{(0)}$ is updated to sparser matrices $\{\hat{\mathbf{A}}_\kappa^{(p)}, \kappa \in \{\text{row}, \text{col}\}\}$. (We omit the description for $\kappa$ in this paragraph for clearer statement). The model is able to roughly detect which edges are not beneficial to the future state prediction, and eliminates these edges in the new representation. As a consequence, any of the matrices in $\{\hat{\mathbf{A}}_\kappa^{(p)}\}$ is sparser than $\mathbf{A}^{(0)}$. Thus the first term in Equation 3 is re-framed as $I(\mathbf{Z}; V^t, \{\hat{\mathbf{A}}_\kappa^{(p)}\})$, which obviously supports: $I(\mathbf{Z}; V^t, \{\hat{\mathbf{A}}_\kappa^{(p)}\}) \leq I(\mathbf{Z}; V^t, \mathbf{A}^{(0)})$, and results in a tighter bound for $\mathbf{Z}$ in Equation 3.

**Relaxation.** Besides that, our iterative training process imposes relaxation to connect the two phases, i.e., label-fitting and representation-compression, of training a deep learning network as a cycle. In the first phase, our model for structural inference tries to fit the target for prediction, when the training errors become small, it turns to the second phase and compresses the representation of any layer to be minimally sufficient to fit the target. In the second phase, the compression process affects the adjacency matrix in latent space as well, and due to the difficulty of training in an unsupervised manner, it is very apt to reach a local optimal. The iterative process, which feeds the learned adjacency matrix back to the encoder, stops the phase of representation-compression, and starts the next round of label-fitting and representation-compression. At this round, the combination of direction information and the initial adjacency matrix together creates a different adjacency matrix for the encoder compared with the previous round, which will relax the learning of adjacency matrix back from the local optimal. We verify the effectiveness of the iterative process through mutual information in Section D.3.

**Direction information.** One of the main differences between undirected and directed graphs is that the former can be described with a unified degree matrix, but the latter requires two matrices for in-degree and out-degree. As a result, we introduce direction information by integrating the in- and out-degrees of every node with the new adjacency matrices, which contributes to a bifurcated training process and encourages the learned embeddings for edges to be more diverse. The integration of direction information is beneficial to the structural inference for directed graphs, which is validated experimentally in Section 5.1.

## 4.5 Training with a Hybrid Loss

Although the iterative process is an effective way to learn the adjacency matrix of directed graphs, adequate loss function can promote the training process as well. Firstly, the following is the loss function for prediction:

$$\mathcal{L}_p = \mathbb{E}_{q_\phi(\mathbf{Z}|V,\mathbf{A})}[\log p_\theta(V \mid \mathbf{Z})] . \tag{17}$$

Then the Kullback–Leibler (KL) divergence acts as a regularization for the latent space:

$$\mathcal{L}_{KL} = -\mathrm{KL}[q_\phi(\mathbf{Z} \mid V)\|p_\theta(\mathbf{Z})] . \tag{18}$$

We omit the superscripts of vectors $V$ and $v$ in Equations 17 - 21, the actual correspondence refers to Sections 4.2 and 4.3.

In addition to the aforementioned terms which are often utilized as the objective function for VAEs, we introduce extra regularization terms. The regularization term of signal *smoothness* takes both inferred structure and node features into consideration to eliminate any connection which does not fulfill the smoothness assumption of adjacent node features, such as indirect interactions. We consider input features $V$ as graph signals, and the assumption for graph signals also works here: values change smoothly across adjacent nodes [10]. Therefore, we introduce Dirichlet energy [3] to measure the smoothness between the signals:

$$\Omega(\mathbf{Z}, V) = \frac{1}{n^2} \sum_{i,j} \mathbf{Z}_{ij} \|v_i - v_j\|^2 . \tag{19}$$

It is worth mentioning that in most cases the adjacency matrix is asymmetric, we cannot use Laplacian matrices to simplify the formulation.

Moreover, we introduce more regularization terms [3, 18] to feature the learned features in latent space with the properties of realistic adjacency matrices, such as *connectiveness* $\mathcal{L}_d$ and *sparsity* $\mathcal{L}_s$. Training the network with these regularization terms encourages our model to further disentangle

Table 1: AUROC values (%) of iSIDG and baselines on synthetic networks

| METHOD | LI | LL | CY | BF | TF | BF-CV |
|--------|----|----|----|----|----|-------|
| iSIDG | **86.2**$\pm$ **4.2** | **88.1**$\pm$ **4.9** | **79.5**$\pm$ **3.8** | **68.3**$\pm$ **7.4** | **60.2**$\pm$ **9.5** | **70.7**$\pm$ **6.0** |
| NRI | 70.5$\pm$ 5.0 | 75.0$\pm$ 4.7 | 64.5$\pm$ 6.2 | 59.0$\pm$ 3.6 | 55.1$\pm$ 7.6 | 59.2$\pm$ 7.4 |
| fNRI | 73.0$\pm$ 5.3 | 77.6$\pm$ 4.9 | 67.1$\pm$ 7.0 | 63.8$\pm$ 6.9 | 59.0$\pm$ 7.1 | 64.8$\pm$ 8.2 |
| MPIR | 46.8$\pm$ 5.6 | 49.0$\pm$ 6.4 | 31.7$\pm$ 10.9 | 47.5$\pm$ 8.0 | 40.9$\pm$ 6.1 | 49.2$\pm$ 4.0 |
| ACD | 65.0$\pm$ 5.2 | 68.4$\pm$ 4.3 | 62.9$\pm$ 3.7 | 59.8$\pm$ 6.7 | 57.2$\pm$ 7.0 | 55.8$\pm$ 6.7 |

node representations from the latent space and distinguish indirect interactions from direct ones:

$$\mathcal{L}_d = -\frac{1}{n}\mathbf{1}^\top \log(\mathbf{Z1}) \text{, and } \mathcal{L}_s = \frac{1}{n^2}\|\mathbf{Z}\|_F^2 \,, \tag{20}$$

where the logarithmic barrier in $\mathcal{L}_d$ forces the model to learn a connected adjacency matrix, but lacks the regularization of sparsity. In consequence, we introduce $\mathcal{L}_s$ to regularize its sparsity.

To conclude, we train our iSIDG with the hybrid loss $\mathcal{L}$:

$$\mathcal{L} = \mathcal{L}_p + \mu \cdot \mathcal{L}_{KL} + \alpha \cdot \Omega(\mathbf{Z}, V) + \beta \cdot \mathcal{L}_d + \gamma \cdot \mathcal{L}_s \,, \tag{21}$$

where $\{\mu, \alpha, \beta, \gamma\}$ are hyperparameters. The first two terms in Equation 21 formulate a weighted ELBO, which is the training objective for VAEs. The rest of the terms are regularization for the learned adjacency matrix in latent space. Together with the training cycles of fitting and compression, which are connected with relaxations, the hybrid loss function further encourages our iSIDG model to reconstruct the adjacency matrix precisely. We give a detailed discussion on the design methodology of iSIDG on how to effectively eliminate indirect connections in Section A.

## 5 Experiments

We test our iSIDG model on three different datasets, and present its ablation study on one of the datasets. Implementation details and further experimental results can be found in Sections B - F.

### 5.1 Structural Inference on Different Systems

**Datasets.** We test our model on the six directed synthetic biological networks [35], namely Linear (LI), Linear Long (LL), Cycle (CY), Bifurcating (BF), Trifurcating (TF), and Bifurcating Converging (BF-CV) networks. These networks are essential components that lead to a variety of different trajectories that are commonly observed in differentiating and developing cells [39]. We use BoolODE [35] to simulate the process of developing cells with these synthetic networks, and obtain six groups of trajectories. The features at every node are one-dimensional, i.e., the level of mRNA expression.

We also test our model on NetSim datasets [43] of simulated fMRI data. The NetSim datasets consist of simulated blood-oxygen-level-dependent imaging data across different regions within the human brain, which leads to asymmetric relation networks. The node features are one-dimensional.

The last datasets to be mentioned here are three physical simulations mentioned in [22], namely springs, charged particles and phase-coupled oscillators (Kuramoto model). We keep the symmetric interactions in the setting of these networks. However, different from the setting mentioned in that work, we sample the trajectories with fixed interactions between the agents in the system, but with different initial conditions, in order to simulate the real application scenario on the observational data from a specific unknown system. The features at every node are 4-dimensional.

**Baselines and metrics.** We compare iSIDG with the state-of-the-art models for structural inference:

- NRI [22]: a VAE-based model for unsupervised relational inference.
- fNRI [51]: an NRI-based model with additional latent space for every factorized interaction type.
- MPIR [52] a model based on minimum predictive information regularization.
- ACD [30]: a variational model that leverages shared dynamics to infer causal relations.

We describe the implementation details of the baseline methods in Section B.2. We demonstrate our evaluation results with the following metrics: the area under the receiver operating characteristic (AUROC), the area under the precision-recall curve (AUPRC) and the Jaccard similarity index (JI). We also provide results of two DAG-based structure learning methods on our datasets in Section E.

Table 2: AUROC values (%) of iSIDG and baselines on physical simulations and NetSim datasets.

| METHOD | Springs | Particles | Kuramoto | NetSim1 | NetSim2 | NetSim3 |
|--------|---------|-----------|----------|---------|---------|---------|
| iSIDG | **90.5**± 3.9 | **78.1**± 4.9 | 79.8± 5.5 | **75.5** ± 5.2 | **72.4** ± 5.2 | **71.0** ± 4.2 |
| NRI | 89.2± 2.6 | 75.1± 2.8 | 78.1± 4.3 | 72.1 ± 3.6 | 72.0 ± 4.1 | 68.2 ± 3.3 |
| fNRI | 90.4± 1.9 | 77.0± 3.0 | 79.5± 5.2 | 72.5 ± 4.6 | 71.2 ± 5.0 | 69.6 ± 3.9 |
| MPIR | 59.5± 3.2 | 55.7± 4.9 | 50.2± 6.3 | 47.2 ± 3.1 | 46.0 ± 3.8 | 44.3 ± 1.7 |
| ACD | 90.1± 4.1 | 77.9± 3.6 | **80.5**± **5.6** | 66.7 ± 3.6 | 64.0 ± 3.1 | 62.9 ± 2.5 |

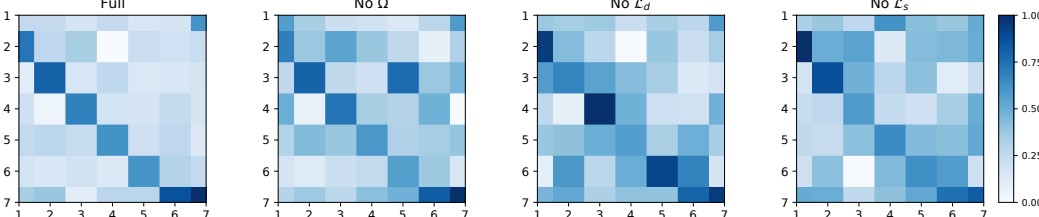

Figure 2: Reconstruction results of iSIDG with different objective functions on LI dataset.

**Results.** The experimental results of our iSIDG model and baseline methods are summarized in Tables 1 and 2 (also Tables 6 - 9 in Appendix), which consist of mean values and 95% intervals of AUROC, AUPRC and JI from 15 experiments on each, respectively. Due to the alternative simulation pipelines between ours and other literature, we obtain different results of ACD and MPIR on physical simulations from other literature. iSIDG outperforms or matches the baseline methods. In the datasets of synthetic networks and three NetSim datasets, which consist of directed graphs and low dimensional input features, iSIDG outperforms baseline models by at most 13.2 percent in terms of AUROC, showing that it can infer structure more precisely than the baseline methods. These results highlight the validity of integrating the iterative training process and the direction information on structural inference problem for directed graphs. It is also remarkable that iSIDG matches or is slightly inferior to the baseline methods in the datasets of physical simulations, especially fNRI and ACD. The reason may be that high dimensional input features (4-dimensional features for physical simulation datasets) contain richer dynamics for fNRI and other methods to infer from. It is worth mentioning that fNRI method utilizes ground truth to distinguish every latent space and match them to the edge type with the highest accuracy, which results in higher AUROC, AUPRC and JI scores. Moreover, since ACD relies on the information shared by input features, the richer information contained in high dimensional features promotes a more promising result.

## 5.2 Ablation Study

We conduct ablation studies on the effectiveness of regularization terms in objective function: signal smoothness $\Omega$, connectiveness $\mathcal{L}_d$ and sparsity $\mathcal{L}_s$, and the reconstruction results are shown in terms of heat maps in Figure 2. The heat maps visualize the average results of 15 experiments of iSIDG with different objective functions by deleting one regularization term, respectively. As shown in the second plot in Figure 2, without considering signal smoothness, the indirect interactions emerge again in the results of structural inference with the correctly inferred edges. The third plot in Figure 2 shows that without considering the connectiveness, the model correctly infers only a few direct interactions but as isolated "islands", and connects these islands with numerous indirect interactions. The last plot in Figure 2 demonstrates the results that without the terms to regularize the sparsity, many direct interactions are substituted by the indirect ones. Therefore, the extra regularization terms in the objective function of iSIDG effectively encourage the model to infer realistic graph structures and to eliminate the influence of indirect interactions. More ablation studies can be found in Section D.4.

## 5.3 Experiments on Scalability and Real-world Datasets

Although iSIDG works well on small datasets from simulations, such as synthetic networks, physical simulations and NetSim datasets, we are curious about the performance of iSIDG on large datasets and real-world datasets. For this reason, we create another "springs" dataset with 100 nodes (we

name it as "Springs100"), and with the same settings as the details mentioned in Section C.3. Besides that, we also test our model and baselines on two gene regulatory network (GRN) datasets: ESC dataset [4] and HSC dataset [32]. As a brief description about the GRNs: ESC has 96 nodes and 409 edges, while HSC has 33 nodes and 126 edges. We ran the iSIDG and baseline methods on three datasets for ten rounds, and summarize the averaged AUROC results in Table 3. As shown in the table, our iSIDG still outperforms baseline methods on all of the three datasets.

Table 3: AUROC values (%) of iSIDG and baselines on "Springs100", ESC and HSC datasets.

| METHOD | Springs100 | ESC | HSC |
|---|---|---|---|
| iSIDG | $\mathbf{72.7}_{\pm \mathbf{6.8}}$ | $\mathbf{56.7}_{\pm \mathbf{5.1}}$ | $\mathbf{58.6}_{\pm \mathbf{4.0}}$ |
| NRI | $67.1_{\pm 5.9}$ | $39.2_{\pm 5.0}$ | $42.9_{\pm 4.3}$ |
| fNRI | $68.4_{\pm 5.5}$ | $39.8_{\pm 4.5}$ | $45.0_{\pm 5.0}$ |
| MPIR | $35.6_{\pm 3.2}$ | $30.6_{\pm 6.2}$ | $37.6_{\pm 3.3}$ |
| ACD | $66.0_{\pm 5.7}$ | $38.7_{\pm 4.2}$ | $44.3_{\pm 4.1}$ |

However, we think that iSIDG cannot prove its scalability in a broader view:

1. Despite the iSIDG performing better than baselines on "Springs100" dataset, the AUROC only values 72.7%, which makes the inferred results to be highly unauthentic for downstream tasks.
2. On ESC dataset, iSIDG can only work better than guessing the existence of edges with equal probability (either exist or not). So that the results are not reliable.
3. Compared with the results shown in Section 5, we observe a significant decline, which suggests that we can extrapolate an even worse inference result on datasets with more nodes.
4. Because the latent space in iSIDG has a fixed size, which means that we have to adopt a larger latent space for larger graphs. Thus the scalability of the iSIDG is limited by VAE as well.

Therefore, we suggest that iSIDG can be utilized to infer the structure of dynamical systems with tens of nodes. We will investigate the problem of scalability in structural inference in our future work, which is also interesting and of practical value.

## 6 Future Work

We so far only consider the structural inference for static and (relatively) small graphs, though we tested the scalability of iSIDG on graphs of 100 nodes. Future research includes introducing dynamic interactions and more scalable model design which remains as a critical challenge in structure learning for dynamical systems. Furthermore, we also want to integrate prior knowledge such as partial structural information to further promote the performance of iSIDG.

## Acknowledgments and Disclosure of Funding

This work is partially supported by the Audacity project GENERIC of Institute for Advanced Studies (IAS) at University of Luxembourg.

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
