# A    Discussion

## A.1    Indirect connections

Originally, iSIDG starts training with a fully connected graph structure: $\mathbf{A}^{(0)}$. Based on the problem setting of dynamical systems, we assume $\mathbf{A}^{(0)}$ can be decomposed as: $\mathbf{A}^{(0)} = \{\mathbf{A}_D, \mathbf{A}_{IN}, \mathbf{A}_U\}$, where $\mathbf{A}_D$ represents the actual edges in the graph (directed connections), $\mathbf{A}_{IN}$ denotes the indirect connections between the two actual connected nodes, and $\mathbf{A}_U$ is the set of non-connections.

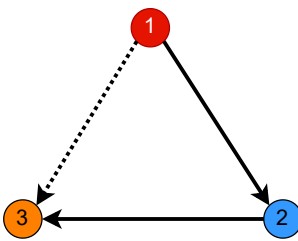

Figure 3: A draft of the direct connections and indirect connections in a three node set. Edges $(v_1, v_2)$ and $(v_2, v_3)$ are direct connections, and edge $(v_1, v_3)$ is the possible indirect connection inferred by structural inference methods.

Previous structural inference methods, such as NRI, fNRI and ACD, are good at eliminating $\mathbf{A}_U$ in the inference results. However, as shown in Figure 3, these methods may falsely reconstruct the structure with indirect connections. It is interesting that the indirect connections resulted from the transmission of signals between nodes. In Figure 3, there exists a chain connection from node $v_1$ to $v_3$ through $v_2$, and in a dynamical system, the changes in $v_1$ will finally affect $v_3$, which would mislead the structural inference methods to falsely infer an edge from $v_1$ to $v_3$. Moreover, for a sparse graph, it is possible that $\mathbf{A}_{IN} \cap \mathbf{A}_U \neq \emptyset$, which challenges structural inference methods, such as NRI, fNRI and ACD.

## A.2    Solution for Eliminating Indirect Connections

Our solution to eliminate indirect connections in the inference results consists of three methodologies: (1) **VAE**; (2) **Iterative process**; and (3) **Regularization terms**.

**VAE.** As shown in previous works [22, 51, 30], the bottleneck structure of VAEs can correctly identify many non-connections in the reconstructed adjacency matrix. Thus, we also rely on the VAE to firstly identify as many non-connections as possible before the first round of iterative process, and sample $\{\hat{\mathbf{R}}_{\text{row}}^{(p)}, \hat{\mathbf{R}}_{\text{col}}^{(p)}\}$ according to operations shown in Equations 11 - 14.

**Iterative process.** The iterative process follows the methodology of VIB, and is mentioned in Section 4.4 with a new objective function:

$$\mathbf{Z} = \arg \min_{\mathbf{Z}} I(\mathbf{Z}; V^t, \{\hat{\mathbf{A}}_\kappa^{(p)}\}) - \mathfrak{u} \cdot I(\mathbf{Z}; V^{t+1}). \tag{22}$$

The iterative process is characterized by a main advantage: the first term in Equation 22 $I(\mathbf{Z}; V^t, \{\hat{\mathbf{A}}_\kappa^{(p)}\}) < I(\mathbf{Z}; V^t, \mathbf{A}^{(0)})$, which results in a tighter bound than Equation 3. In other words, after feeding the direction information back to the input side of the encoder, the new search space excludes the non-connections, which are found by VAE in the previous training epochs. With a smaller search space, iSIDG can focus on the differentiation between misleading indirect connections and the real connections.

**Regularization terms.** Although feeding the learned adjacency matrix back to the encoder can eliminate the indirect connections, we need regularization terms to secure and accelerate this process. The sparsity term $\mathcal{L}_S$ encourages iSIDG to choose the more solid path from every node, to meet the minimal sufficient statistics assumption of IB. For example, in Figure 3, at node 1, iSIDG has to choose between the path $1 \rightarrow 3$ and $1 \rightarrow 2 \rightarrow 3$ based on the sparsity regularization terms and the minimal sufficient information for dynamics prediction in decoder.

One may argue that it is also possible that iSIDG will falsely think there are edges from node 1 to node 2, and from node 1 to node 3. However, this does not conform to the future state prediction. The task of the decoder is trying to approximate:

$$v_i^{t+1} = v_i^t + \Delta \cdot \sum_{j \in \mathcal{N}_i} f\left(||v_j^t, v_i^t||_\alpha\right), \tag{23}$$

where $f(\cdot)$ denotes the interaction from node $v_j$ to node $v_i$. We may imagine the interaction could be RNA activation level in GRN, spring force in spring-balls system, and the phase-couple of oscillators. As a result, the interactions between nodes are assumed to be the same, and described with the same function. Yet node 1 can only affect node 3 through node 2, which results in a superposition of functions: $f(f(\cdot))$. Therefore, both edges from node 1 to node 2, and from node 1 to node 3 cannot predict the future state of node 3 correctly, and VAE will search for other possible edges.

## B    Further Implementation Details

### B.1    Implementation details of iSIDG

We summarize the described architecture of iSIDG and present the pipeline of training iSIDG in Algorithm 1.

---

**Algorithm 1** General Framework for iSIDG

1: **Input:** $V$, number of nodes $n$
2: **Parameters:** $\delta$, $\sigma$, $\eta$, $\mu$, $\alpha$, $\beta$, $\gamma$, $\xi$, batch size $m$ and time steps $T$
3: **Model Weights:** $\phi, \theta$
4: **Output:** $\mathbf{A}^{(t)}$
5: $i \leftarrow 1$
6: $StopCond \leftarrow \|\mathbf{A}^{[i]} - \mathbf{A}^{[i-1]}\|_F^2 < \delta\|\mathbf{A}^{(0)}\|_F^2$
7: $\mathbf{A}^{(0)} = \mathbb{1} \in n \times n$
8: $\mathbf{A}^{(i+1)} = \mathbf{A}^{(0)}$
9: Split $V$ into $\mathbf{V}^T = \{V^0, \dots, V^T\}$ and $\mathcal{V}^{T+1} = \{V^1, \dots, V^{T+1}\}$
10: **while** $i < MaxEpoch$ **do**
11:      $\mathbf{Z} \leftarrow \text{Encoder}(\mathbf{V}^T, \mathbf{A}^{(i)}, \phi)$
12:      $\hat{\mathbf{V}}^{T+1} \leftarrow \text{Decoder}(\mathbf{V}^T, \mathbf{Z}, \theta)$
13:      **if** $i > \eta$ and $StopCond$ **then**
14:          $\tilde{\mathbf{R}}_\kappa^{(i)} \leftarrow \{\mathbf{Z}, \mathbf{A}^{(0)}, \xi\}$ using Eqs. 11 - 15
15:          $\mathbf{A}^{(i)} \leftarrow \tilde{\mathbf{R}}_\kappa^{(i)}$
16:          $\mathcal{L}_p \leftarrow \{\hat{\mathbf{V}}^{T+1}, \mathbf{V}^{T+1}, \delta\}$
17:          $\mathcal{L}_{KL}$, $\mathcal{L}_d$, $\mathcal{L}_s \leftarrow \{\mathbf{Z}, n\}$
18:          $\mathcal{L}^{(i)} \leftarrow \mathcal{L}_p + \mu\mathcal{L}_{KL} + \alpha\Omega(\mathbf{A}^{(t)}, V) + \beta\mathcal{L}_d + \gamma\mathcal{L}_s$
19:          $\mathcal{L} \leftarrow \mathcal{L}^{(i)}/T$
20:          Back-propagate $\mathcal{L}$ to update model weights $\phi$ and $\theta$
21:          $i \leftarrow i + 1$
22:      **end if**
23: **end while**
24: **Return:** $\mathbf{A}^{(t)}$

---

**Algorithm 2** A MLP block.

1: **Input:** features $input$
2: x = elu(Linear1($input$))
3: x = dropout(x)
4: x = elu(Linear2(x))
5: out = batch_norm(x)
6: **Return:** out

---

**Algorithm 3** Pseudocode for MPM encoder.

1: **Input:** features $input$
2: x = mlp1($input$)
3: x = node2edge(x)
4: x = mlp2(x)
5: x_skip = x
6: x = edge2node(x)
7: x = mlp3(x)
8: x = node2edge(x)
9: x = concatenation([x, x_skip])
10: x = mlp4(x)
11: out = fully connected out(x)
12: **Return:** out

---

**Basic settings.** We implement our iSIDG model in PyTorch [33] with the help of Scikit-Learn [34] to calculate various metrics. We run experiments on a single NVIDIA Tesla V100 SXM2 graphic card, which has 32 GB graphic memory and 5120 NVIDIA CUDA Cores. During training, we set batch size as 128 for datasets which have less than 10 nodes, for those more than 10 nodes, we set batch size as

64. We train our iSIDG model with 2000 epochs on every dataset, which takes around 30 - 42 hours for every training session. The code can be found at `https://github.com/AoranWANGRalf/iSIDG`.

**Loss Functions.** The hybrid loss function shown in Equation 21 consists of five different terms to be calculated: $\{\mathcal{L}_p, \mathcal{L}_{KL}, \Omega(\mathbf{Z}, V), \mathcal{L}_d, \mathcal{L}_s\}$. We would like to describe the actual implementations of $\mathcal{L}_p$ and $\mathcal{L}_{KL}$. Since the target of the decoder is the prediction of future node features based on present states, $\mathcal{L}_p$ is estimated by:

$$-\sum_j \sum_{t=2}^{T} \frac{\|v_j^t - \hat{\mathbf{h}}_j^t\|^2}{2\sigma^2} + \text{const} ,\tag{24}$$

where the variance $\sigma$ is set as a fixed value, and the const is $e^{-16}$.

And the calculation of KL-divergence $\mathcal{L}_{KL}$ over a uniform prior can be simplified as the calculation of entropy $H$:

$$\sum_{i,j} H(q_\theta(z_{ij}|v)) + \text{const} .\tag{25}$$

**Hyper parameters.** The choice of most hyper parameters in our iSIDG model (such as those mentioned in Algorithm 1) may refer to Table 4. In addition to the hyper parameters mentioned in the table, we set the combination coefficient $\lambda$ in Equation 13 as 0.3 for the training on all of the datasets. We argue that $\lambda$ can be set as varying values according to the training epochs, and decrease during training, which may be helpful for faster converge. We would like to carry on more research onto this hypothesis in the future. Besides that, we describe the choice of time step length $T$ in Section C. Time step length $T$ decides the amount of samplings along a trajectory.

Table 4: Hyper parameter choices for every dataset.

| DATASET | $\delta$ | $\sigma$ | $\eta$ | $\mu$ | $\alpha$ | $\beta$ | $\gamma$ | $\xi$ |
|---|---|---|---|---|---|---|---|---|
| LI | 0.00001 | 0.00008 | 150 | 200 | 50 | 10 | 20 | 0.5 |
| LL | 0.00001 | 0.00008 | 150 | 400 | 80 | 20 | 30 | 0.5 |
| CY | 0.00001 | 0.00008 | 150 | 300 | 50 | 10 | 20 | 0.5 |
| BF | 0.00001 | 0.00008 | 150 | 400 | 80 | 20 | 40 | 0.4 |
| TF | 0.00001 | 0.00008 | 150 | 500 | 70 | 20 | 50 | 0.4 |
| BF-CV | 0.00001 | 0.00008 | 150 | 400 | 80 | 20 | 40 | 0.4 |
| NetSim1 | 0.00001 | 0.00008 | 180 | 300 | 50 | 10 | 20 | 0.5 |
| NetSim2 | 0.00001 | 0.00008 | 180 | 400 | 70 | 20 | 30 | 0.5 |
| NetSim3 | 0.00001 | 0.00008 | 180 | 500 | 80 | 30 | 50 | 0.5 |
| Springs | 0.0001 | 0.00005 | 100 | 100 | 50 | 10 | 20 | 0.5 |
| Particles | 0.0001 | 0.00005 | 100 | 100 | 50 | 10 | 20 | 0.5 |
| Kuramoto | 0.0001 | 0.00005 | 100 | 200 | 50 | 10 | 20 | 0.5 |

**Ablation study with different GNNs.** The main difference between the utilization of different GNNs takes place in the encoder of iSIDG. For the implementation of **MLP** encoder, we use three stacked multi-layer perceptron blocks, with each block consists of the units shown in Algorithm 2. We place a concatenation operation between Block 1 and 2 to get edge representations. The last block is attached with another Linear layer with its dimension matching the number of edges in a full-graph setting.

Table 5: Number of units in the Linear layers in MLP encoder.

| BLOCK | Linear 1 | Linear 2 |
|---|---|---|
| 1 | 256 | 256 |
| 2 | 512 | 256 |
| 3 | 256 | 256 |

The number of units in the Linear layers in the **MLP** encoder is shown in Table 5. The dropout rates in all of the blocks are set as 0.5.

For the simple message-passing mechanism encoder (**MPM** encoder), we follow the design of "MLP Encoder" in NRI [22]. And can be summarized as the pseudocode shown in Algorithm 3.

| **Algorithm 4** Pseudocode for GIN encoder. | **Algorithm 5** Pseudocode for GAT encoder. |
|---|---|
| 1: **Input:** features $input$ | 1: **Input:** features $input$ |
| 2: x = mlp1($input$) | 2: x = mlp1($input$) |
| 3: x_skip = node2edge(x) | 3: x_skip = node2edge(x) |
| 4: x_skip = mlp2(x_skip) | 4: x_skip = mlp2(x_skip) |
| 5: x = $f_{GIN}$(x) | 5: x = $f_{GAT}$(x) |
| 6: x = mlp3(x) | 6: x = mlp3(x) |
| 7: x = node2edge(x) | 7: x = node2edge(x) |
| 8: x = concatenation([x, x_skip]) | 8: x = concatenation([x, x_skip]) |
| 9: x = mlp4(x) | 9: x = mlp4(x) |
| 10: out = fully connected out(x) | 10: out = fully connected out(x) |
| 11: **Return:** out | 11: **Return:** out |

Based on the design of Graph Identity Networks and the **MPM** encoder, we combine these idea together into the **GIN** encoder, by replacing the operations from line 3 to line 6 in Algorithm 3 with the GIN operation mentioned in [53], but still keep the skip connection for edges representations, and the pseudocode is shown in Algorithm 4. Similarly, if we replace the GIN operation in line 5 of Algorithm 4 with GCN operation, we obtain the **GCN** encoder. Moreover, the design of **GAT** encoder is similar to that of **GIN** encoder and is shown as the pseudocode shown in Algorithm 5.

## B.2   Implementation details of baselines

**NRI.** We use the official implementation code by the author from `https://github.com/ethanfetaya/NRI` with customized data loader for our chosen datasets: synthetic networks, Net-Sims and physical simulations. We add our metric-evaluation in "test" function, after the calculation of accuracy in the original code.

**fNRI.** We use the official implementation code by the author from `https://github.com/ekwebb/fNRI` with customized data loader for our chosen three datasets. We add our metric-evaluation in "test" function, after the calculation of accuracy and the selection of correct order for the representations in latent spaces in the original code.

**MPIR.** We use the official implementation code from `https://github.com/tailintalent/causal` as the code for MPIR. We run the code by customized data loader for the chosen three datasets. After the obtain of the results, we run another script to calculate the metrics.

**ACD.** We follow the official implementation code by the author as the framework for ACD (`https://github.com/loeweX/AmortizedCausalDiscovery`). We run the code with customized data loader for the chosen three datasets. We implement the metric-calculation pipeline in the "forward_pass_and_eval()" function.

## C   Further Details about Datasets

### C.1   Synthetic networks

The six directed Boolean networks (LI, LL, CY, BF, TF, BF-CV) are the most often observed fragments in many gene regulatory networks, each has 7, 18, 6, 7, 8 and 10 nodes, respectively. Thus by carrying out experiments on these networks, we can get acknowledge about the performance of the chosen methods on the structural inference of real-world biological networks. We collect the six ground-truth directed Boolean networks from [35] and simulate the single cell evolving trajectories with BoolODE [35] `https://github.com/Murali-group/BoolODE` with default settings mentioned in that paper for every network. We sample 12000 trajectories and group them into three datasets: for training, for validation, and for testing, each with the number of 8000, 2000 and 2000, respectively. Then we sample 49 snapshots with equal time interval in every trajectory and save them as ".npy" files for data loading.

## C.2 NetSim datasets

The NetSim datasets simulate blood-oxygen-level-dependent imaging data across different regions within the human brain and is described in [43] and `https://www.fmrib.ox.ac.uk/datasets/netsim/`. We target at inferring the existence of directed connections between different brain areas. Among the total 28 datasets in NetSim, we choose the first three datasets (NetSim1, NetSim2 and NetSim3) which have 5, 10, and 15 nodes, respectively. We sample 49 snapshots on each trajectory with equal interval and randomly group them into three sets for training, validation and testing with the ratio of 8: 2: 2, respectively.

## C.3 Physical simulations

To generate these physical simulations (springs, charged particles and phase-coupled oscillators), we follow the description of the data in [22] but with fixed interactions. To be specific, at the beginning of the data generation for each physical simulation, we randomly generate a ground truth graph and then simulate 12000 trajectories on the same ground truth graph, but with different initial conditions. The rest settings for the simulations are the same as that mentioned in [22]. It is worth mentioning that the connections in the physical simulations are indirected, which are different from those in the synthetic networks and NetSim datasets. We collect the trajectories and randomly group them into three sets for training, validation and testing with the ratio of 8: 2: 2, respectively.

# D  Further Details about Experiments

## D.1 Metrics

We choose the three most representative metrics from 0-1 classification for the evaluation of structural inference results, namely the area under the receiver operating characteristic (AUROC), the area under the precision-recall curve (AUPRC) and Jaccard index (JI). All of the metrics are calculated from the inferred structure and the full-patch of ground-truth.

**AUROC.** The area under the receiver operating characteristic (AUROC) is a performance metric to evaluate the result of classification tasks. It demonstrates the model's ability to discriminate between cases (positive examples) and non-cases (negative examples). For our case, it is used to make clear the model's ability to distinguish actual edges and empty interactions. It is calculated as the area under the receiver operating characteristic curve (ROC), which sets false positive rate (FPR) as x-axis and true positive rate (TPR) as y-axis at various threshold settings. In the ROC space, the best possible prediction method would yield a point in the upper left corner or coordinate (0,1) of the ROC space, representing 100% no false negatives and 100% no false positives. Then a model outputs more precise results with many TPR yields a higher AUROC value, and vice versa.

**AUPRC.** The area under the precision-recall curve (AUPRC) is a useful performance metric for imbalanced data in a problem setting since we care a lot about finding the existence of interactions between nodes. It shows the ability of the model to find the existence of all ground-truth edges without accidentally marking any non-interaction as existence. The AUPRC is calculated as the area under the Precision-recall (PR) curve. The x-axis of a PR curve is the TPR and the y-axis is the precision. This is in contrast to ROC curves, where the y-axis is the TPR and the x-axis is FPR. Similar to calculate AUROC, AUPRC is also calculated from various threshold settings.

**Jaccard index.** The Jaccard index (sometimes called the Jaccard similarity coefficient) (JI) compares the inferred edges and ground-truth to see the shared results and the distinct results. Jaccard index is a measure of similarity for the two sets of data, with a range from 0% to 100%. The higher the percentage, the more similar the two populations. It is calculated as "Intersection of two sets/Union of two sets". Although JI is easy to calculate and interpret, the result is extremely sensitive to small samples sizes and may produce erroneous results, especially when the number of samples is small or the data sets contain missing observations.

Table 6: AUPRC values (%) of iSIDG and baselines on synthetic networks.

| METHOD | LI | LL | CY | BF | TF | BF-CV |
|---|---|---|---|---|---|---|
| iSIDG | **40.3**$_{\pm 3.4}$ | **18.3**$_{\pm 2.9}$ | **42.1**$_{\pm 5.7}$ | **38.0**$_{\pm 5.0}$ | **46.6**$_{\pm 5.6}$ | **32.8**$_{\pm 4.4}$ |
| NRI | 27.3$_{\pm 3.1}$ | 7.30$_{\pm 1.3}$ | 31.1$_{\pm 4.7}$ | 28.2$_{\pm 3.1}$ | 32.7$_{\pm 3.6}$ | 17.1$_{\pm 3.6}$ |
| fNRI | 29.0$_{\pm 3.4}$ | 8.59$_{\pm 1.9}$ | 25.8$_{\pm 5.7}$ | 29.8$_{\pm 4.3}$ | 33.2$_{\pm 4.9}$ | 20.1$_{\pm 3.5}$ |
| MPIR | 15.6$_{\pm 2.6}$ | 5.78$_{\pm 1.4}$ | 16.7$_{\pm 2.5}$ | 23.7$_{\pm 3.3}$ | 31.3$_{\pm 2.1}$ | 17.8$_{\pm 2.0}$ |
| ACD | 21.9$_{\pm 3.5}$ | 14.7$_{\pm 2.5}$ | 29.9$_{\pm 5.4}$ | 23.2$_{\pm 4.8}$ | 34.3$_{\pm 4.0}$ | 24.8$_{\pm 3.6}$ |

Table 7: JI values (%) of iSIDG and baselines on synthetic networks.

| METHOD | LI | LL | CY | BF | TF | BF-CV |
|---|---|---|---|---|---|---|
| iSIDG | **45.4**$_{\pm 3.9}$ | **49.5**$_{\pm 5.0}$ | **47.2**$_{\pm 5.5}$ | **46.8**$_{\pm 4.8}$ | **46.0**$_{\pm 4.3}$ | **46.6**$_{\pm 3.6}$ |
| NRI | 41.6$_{\pm 2.9}$ | 37.2$_{\pm 4.4}$ | 36.5$_{\pm 4.8}$ | 38.0$_{\pm 4.1}$ | 35.9$_{\pm 4.3}$ | 40.6$_{\pm 3.5}$ |
| fNRI | 41.8$_{\pm 3.0}$ | 38.5$_{\pm 5.7}$ | 33.5$_{\pm 6.2}$ | 32.4$_{\pm 5.1}$ | 35.0$_{\pm 5.0}$ | 37.4$_{\pm 3.4}$ |
| MPIR | 11.9$_{\pm 1.7}$ | 3.70$_{\pm 0.8}$ | 13.3$_{\pm 1.4}$ | 18.4$_{\pm 2.6}$ | 24.5$_{\pm 2.0}$ | 15.6$_{\pm 1.6}$ |
| ACD | 22.6$_{\pm 4.5}$ | 11.0$_{\pm 2.6}$ | 20.2$_{\pm 2.8}$ | 15.0$_{\pm 2.2}$ | 18.1$_{\pm 2.5}$ | 14.0$_{\pm 2.8}$ |

Table 8: AUPRC and JI values (%) of iSIDG and baselines on NetSim datasets.

| | AUPRC | | | JI | | |
|---|---|---|---|---|---|---|
| METHOD | NetSim1 | NetSim2 | NetSim3 | NetSim1 | NetSim2 | NetSim3 |
| iSIDG | **40.0**$_{\pm 4.9}$ | **36.3**$_{\pm 4.5}$ | **34.8**$_{\pm 5.0}$ | **49.1**$_{\pm 4.8}$ | **47.6**$_{\pm 5.0}$ | **44.2**$_{\pm 4.5}$ |
| NRI | 34.5$_{\pm 2.9}$ | 32.4$_{\pm 3.0}$ | 32.1$_{\pm 3.3}$ | 43.9$_{\pm 3.5}$ | 42.3$_{\pm 2.8}$ | 41.2$_{\pm 2.9}$ |
| fNRI | 33.1$_{\pm 3.7}$ | 30.8$_{\pm 3.8}$ | 30.2$_{\pm 4.1}$ | 44.1$_{\pm 4.2}$ | 40.9$_{\pm 4.0}$ | 41.7$_{\pm 3.6}$ |
| MPIR | 25.2$_{\pm 2.2}$ | 23.7$_{\pm 3.1}$ | 21.9$_{\pm 3.0}$ | 25.4$_{\pm 1.3}$ | 24.1$_{\pm 2.6}$ | 23.5$_{\pm 2.9}$ |
| ACD | 32.9$_{\pm 2.6}$ | 30.7$_{\pm 3.5}$ | 30.2$_{\pm 2.1}$ | 26.8$_{\pm 1.7}$ | 25.6$_{\pm 2.4}$ | 23.7$_{\pm 3.5}$ |

Table 9: AUPRC and JI values (%) of iSIDG and baselines on physical simulations.

| | AUPRC | | | JI | | |
|---|---|---|---|---|---|---|
| METHOD | Springs | Particles | Kuramoto | Springs | Particles | Kuramoto |
| iSIDG | **80.7**$_{\pm 6.7}$ | 70.6$_{\pm 4.9}$ | 71.8$_{\pm 5.4}$ | 81.8$_{\pm 4.2}$ | 72.6$_{\pm 4.1}$ | 71.3$_{\pm 3.7}$ |
| NRI | 79.8$_{\pm 4.4}$ | 69.4$_{\pm 2.1}$ | 71.7$_{\pm 4.6}$ | 80.4$_{\pm 2.8}$ | 72.2$_{\pm 2.6}$ | 71.3$_{\pm 3.3}$ |
| fNRI | **80.7**$_{\pm 5.7}$ | **72.6**$_{\pm 3.2}$ | **73.9**$_{\pm 6.8}$ | **82.2**$_{\pm 2.8}$ | **74.6**$_{\pm 2.4}$ | **73.4**$_{\pm 3.7}$ |
| MPIR | 35.5$_{\pm 5.2}$ | 32.4$_{\pm 3.9}$ | 39.0$_{\pm 3.4}$ | 33.7$_{\pm 4.8}$ | 30.4$_{\pm 4.2}$ | 37.3$_{\pm 3.1}$ |
| ACD | 80.2$_{\pm 6.1}$ | 58.4$_{\pm 3.6}$ | 70.4$_{\pm 5.7}$ | 80.0$_{\pm 2.6}$ | 57.3$_{\pm 4.7}$ | 69.4$_{\pm 5.8}$ |

## D.2 Further experimental results

In this section, we demonstrate additional experimental results as the supplement to Section 5. We present AUPRC and JI values as well as a case study about the structural inference results of iSIDG and baseline methods.

**AUPRC and JI results.** We present the AUPRC and JI results of iSIDG and baseline methods on synthetic networks, NetSim datasets and physical simulations in Tables 6 - 9. The results are the mean values and 95% intervals of 15 experiments, respectively. Similar to the AUROC results presented in Section 5.1, iSIDG performs the best among all of the chosen methods in datasets with directed graphs, and can match the best baseline method in datasets with indirected graphs.

Besides that, we conduct a case study of iSIDG with different objective functions, and calculate the average accumulated path lengths based on 15 experiments of each on Linear network dataset. The results are shown in Figure 4. It is clear that our choice of objective function with regularization terms of smoothness, connectiveness and sparsity successfully encourages iSIDG model to infer the structure of the system with less indirect interactions and more accurate results.

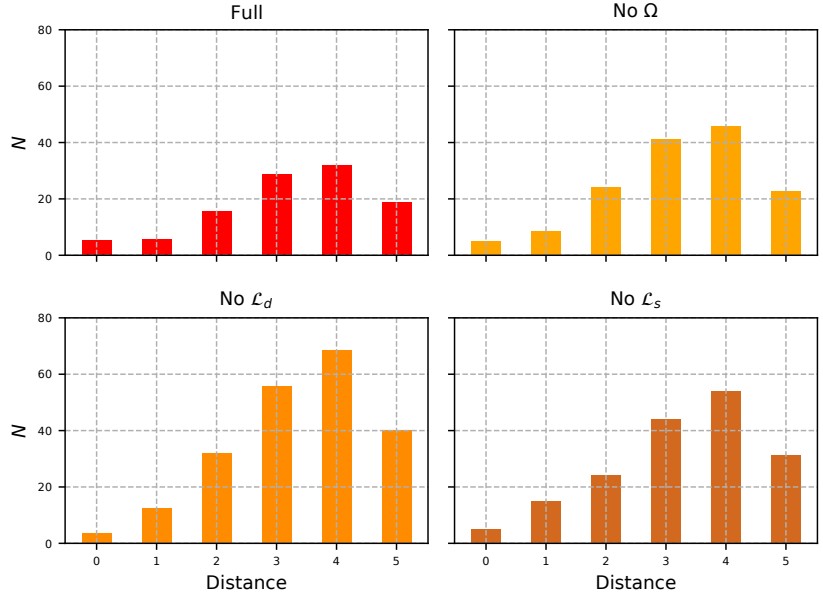

Figure 4: Averaged accumulated length of the connections between nodes of the inference results on Linear network dataset with different objective functions. The y-axis represents the averaged counts of paths according to their length, and the x-axis denotes the path length between two nodes. The results are the mean values of 15 experiments on Linear network dataset with different objective functions.

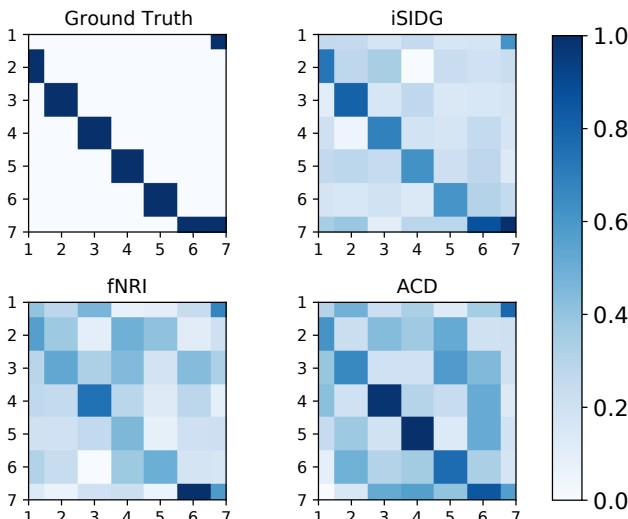

Figure 5: Reconstruction results of different methods on Linear network dataset.

**Case study.** To gain an intuitive understanding of the results of the structural inference of directed graphs, we conduct a case study on the Linear network dataset. The reconstructed adjacency matrices of our iSIDG, fNRI and ACD methods on Linear network dataset are visualized in Figure 5, as well as the ground-truth. The results are the mean results of 15 experiments of each method and are shown in terms of distributions of reconstructed relations. In the figure, $0.0$ at a point $[x, y]$ denotes the absence of interaction from $x$ to $y$, and $1.0$ represents in an opposite way. As shown in Figure 5, compared with the ground-truth, the structure reconstructed by ACD contains many indirect

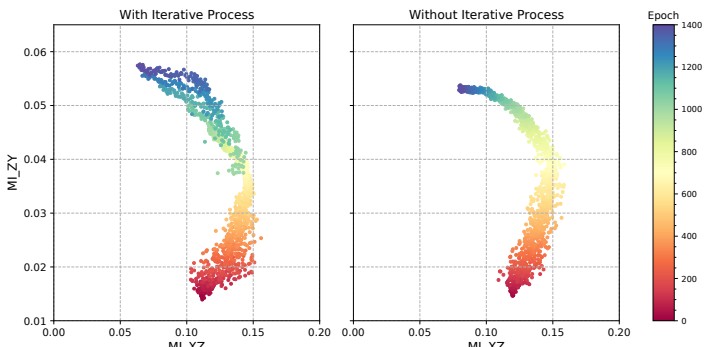

Figure 6: Mutual information comparison between methods with or without iterative process on Linear network dataset.

connections. It is possibly the result from the fact that ACD infers a latent posterior graph from the input features based on hidden confounding [50]. For Linear network dataset, which has a "chain" structure, it may confuse ACD method as it contains many indirect interactions, and thus leads to inaccurate inference results. Moreover, fNRI utilizes a factorized latent space for every possible interaction types. For the case here, it even reserves a latent space for "Non-interaction" type, which further disentangles possible indirect interactions from direct ones, and therefore slightly promotes its accuracy. In contrast, the iterative training process in iSIDG creates relaxation of the gradients during training. As a result, it can distinguish most indirect interactions from the direct ones and reconstruct the adjacency matrix more precisely than the other methods on Linear network dataset.

### D.3 Mutual information

We conduct another case study to demonstrate the difference of the mutual information between our iSIDG with and without the iterative process on Linear network dataset with 10 experiments of every model with 1400 training epochs. We utilize nearest-neighbor MI estimator [36] and the plots are shown in Figure 6. The plots show the mutual information between input features and embeddings in latent space MI_XZ for every edge, as well as the mutual information between embeddings in latent space and the output features MI_ZY for every edge. Since the input consists of node features, we concatenate them according to [sender, receiver] to represent each edge for the calculation of mutual information, respectively.

As shown in Figure 6, in both methods, the two types of mutual information increase in the first phase, and then after around 800 epochs of training, the mutual information between embeddings in latent space and output features decreases, while the other one keeps growing. These phenomena verify the two procedures in the training of deep neural networks as mentioned in the IB theory [42, 1], namely label-fitting and representation-compression.

In the phase of representation-compression, when the stop condition is reached, the method with the iterative process stops the current training round, feeds the obtained adjacency matrix from latent space back to the input side of encoder, and starts another round of training with the obtained adjacency matrix. The phase-transition is demonstrated by the decrease of both mutual information and start with another round of label-fitting and representation-compression. Therefore, the iterative training process creates several rounds of relaxation during training, and encourages our iSIDG model to learn a more compressed representation in the latent space. As shown in Figure 5, the compressed representation in the latent space eliminates most of the indirect interactions and promotes the structural inference accuracy of our iSIDG model.

### D.4 Ablation studies on different encoders and decoders

We conduct another series of ablation studies to demonstrate the different choices of encoders and decoders in our iSIDG model. We choose four methods as encoders for our ablation study,

Table 10: AUROC values (%) of iSIDG with different encoders and decoders on LI dataset.

|  | MLP | MULTI | RNN |
|---|---|---|---|
| MLP | $64.8_{\pm 4.9}$ | $60.6_{\pm 4.9}$ | $60.9_{\pm 4.8}$ |
| MPM | $75.5_{\pm 3.3}$ | $70.4_{\pm 5.4}$ | $71.6_{\pm 3.8}$ |
| GCN | $81.0_{\pm 4.7}$ | $76.5_{\pm 4.0}$ | $76.3_{\pm 4.6}$ |
| GIN | $\mathbf{86.2_{\pm 4.2}}$ | $79.8_{\pm 4.4}$ | $79.5_{\pm 4.8}$ |
| GAT | $65.9_{\pm 3.8}$ | $63.0_{\pm 4.3}$ | $62.7_{\pm 3.9}$ |

Table 11: AUROC values (%) of DAG-GNN and DAG-NoCurl on synthetic networks

| METHOD | LI | LL | CY | BF | TF | BF-CV |
|---|---|---|---|---|---|---|
| DAG-GNN | $42.3_{\pm 2.0}$ | $40.5_{\pm 3.1}$ | $42.9_{\pm 4.5}$ | $43.2_{\pm 3.4}$ | $40.8_{\pm 2.8}$ | $40.5_{\pm 3.7}$ |
| DAG-GNN* | $40.0_{\pm 5.1}$ | $46.8_{\pm 3.6}$ | $44.0_{\pm 5.7}$ | $50.3_{\pm 4.5}$ | $50.2_{\pm 3.6}$ | $51.7_{\pm 4.1}$ |
| DAG-NoCurl | $43.0_{\pm 2.7}$ | $43.2_{\pm 3.3}$ | $49.5_{\pm 4.4}$ | $51.8_{\pm 3.2}$ | $39.4_{\pm 5.5}$ | $51.1_{\pm 3.2}$ |

Table 12: AUROC (%) of DAG-GNN and DAG-NoCurl on physical simulations and NetSim datasets.

| METHOD | Springs | Particles | Kuramoto | NetSim1 | NetSim2 | NetSim3 |
|---|---|---|---|---|---|---|
| DAG-GNN | $48.2_{\pm 2.3}$ | $40.0_{\pm 3.4}$ | $45.5_{\pm 4.5}$ | $43.1_{\pm 3.4}$ | $42.4_{\pm 4.9}$ | $40.9_{\pm 3.6}$ |
| DAG-GNN* | $39.5_{\pm 2.7}$ | $38.4_{\pm 2.1}$ | $38.5_{\pm 3.3}$ | $35.4_{\pm 3.4}$ | $34.0_{\pm 4.6}$ | $33.2_{\pm 3.9}$ |
| DAG-NoCurl | $47.4_{\pm 2.6}$ | $44.8_{\pm 3.6}$ | $47.0_{\pm 4.7}$ | $49.5_{\pm 3.1}$ | $47.2_{\pm 3.6}$ | $46.5_{\pm 3.3}$ |

namely: multi-layer-perceptrons (MLP), simple message-passing mechanism (MPM), GCN, GIN, and GAT. We also take three different decoders into account: MLP, avoiding degenerate decoder (MULTI) [22][51], and recurrent decoder (RNN) [22]. The results are shown in Table 10, where the rows denote the choice of encoder, the columns for decoder, and the results are the average values and 95% intervals of 15 experiments with every combination of encoder and decoders. As shown in Table 10, the methods with GNNs as encoder are greatly more accurate than the ones with simple MLP encoder. The only exception are the ones with GAT encoder, which are only marginally superior to the ones with MLP encoder. The reason may be that GAT is prone to over-fitting and the attention weights lack supervision in the cases of structural inference. When it comes to GIN as encoder, the results are superior to methods with other encoders by at least 3.2 percent of AUROC. Surprisingly, methods with MLP decoder, which is the simplest decoder among the three, output the most accurate reconstruction results. We argue that it may be due to the simple task of the iSIDG: only focusing on structural inference instead of both structural inference and dynamic prediction (as that of NRI).

# E  Do DAG-structure Learning Methods Work in Our Problem Setting?

We observe that there is a series of works, which deal with the directed acyclic graphs (DAG) structure learning [59, 57, 58]. These works formulate the problem as a continuous optimization with a structural constraint that ensures acyclicity [59], with explicit structural constraints [57], or with implicit mechanism to force the acyclicity [58]. Although it is claimed by the authors that these work deal with DAG structure learning, which is different from our problem setting, we are still curious about the performance of these works on the problem of structural inference of dynamical systems.

We choose DAG-GNN [57] and DAG-NoCurl [58], which are the representative in the research field of DAG structure learning. We follow the official implementations of these models:

- DAG-GNN: `https://github.com/fishmoon1234/DAG-GNN`;
- DAG-NoCurl: `https://github.com/fishmoon1234/DAG-NoCurl`;

and we only change the data loader modules to load physical simulations, synthetic networks and NetSims datasets, respectively. We ran the experiments for ten rounds, and summarize the AUROC results in Tables 11 and 12, where we note DAG-GNN* as the DAG-GNN with acyclicity constraint set as zero. As shown in the tables, DAG-GNN, DAG-GNN* and DAG-NoCurl fail to infer the existence of interactions in dynamical systems. The reason may be that the existence of cycles in the

Table 13: Training time (in hour) of iSIDG and baseline methods on synthetic networks.

| METHOD | LI | LL | CY | BF | TF | BF-CV |
|--------|------|------|------|------|------|-------|
| iSIDG | 48.2 | 50.6 | 40.8 | 44.7 | 40.3 | 44.0 |
| NRI | 14.3 | 18.2 | 13.0 | 15.5 | 13.6 | 16.9 |
| fNRI | 15.5 | 21.9 | 14.9 | 18.6 | 13.7 | 18.0 |
| MPIR | 5.0 | 14.4 | 3.6 | 8.0 | 5.5 | 7.9 |
| ACD | 40.5 | 42.8 | 39.6 | 44.0 | 41.7 | 43.2 |

Table 14: Training time (in hour) of iSIDG and baseline methods on physical simulations and NetSim datasets.

| METHOD | Springs | Particles | Kuramoto | NetSim1 | NetSim2 | NetSim3 |
|--------|---------|-----------|----------|---------|---------|---------|
| iSIDG | 42.2 | 36.0 | 39.2 | 20.7 | 36.9 | 50.8 |
| NRI | 20.1 | 20.3 | 19.8 | 8.8 | 16.0 | 21.5 |
| fNRI | 22.8 | 20.7 | 19.0 | 9.0 | 17.8 | 25.6 |
| MPIR | 7.9 | 6.6 | 6.3 | 2.1 | 5.6 | 9.5 |
| ACD | 39.8 | 36.4 | 38.0 | 20.5 | 35.8 | 45.7 |

Table 15: Training time (in hour) of iSIDG and baseline methods on "Springs100", ESC and HSC datasets.

| METHOD | Springs100 | ESC | HSC |
|--------|------------|------|------|
| iSIDG | 106.5 | 96.8 | 50.3 |
| NRI | 40.6 | 39.4 | 30.4 |
| fNRI | 49.0 | 42.0 | 31.8 |
| MPIR | 20.7 | 19.5 | 12.0 |
| ACD | 82.4 | 80.4 | 51.8 |

Table 16: Counts of total parameters (in million) of iSIDG and baseline methods.

| | |
|-------|------|
| iSIDG | 1.72 |
| NRI | 1.12 |
| fNRI | 1.12 |
| MPIR | 0.62 |
| ACD | 3.70 |

systems, which is common observed among dynamical systems, violates the acyclicity assumptions of these methods. In particularly, although the acyclicity constraint in DAG-GNN* is set as zero, DAG-GNN* is unable to correctly infer the structure of our datasets, and (un)surprisingly performs even worse than DAG-GNN.

## F   Time and Memory Efficiency

We summarized the training time of iSIDG and baseline methods on all of the datasets mentioned in this work in Tables 13 - 15. All of the results are the averaged training time of 10 rounds of each method, and is summarized in hours. Unfortunately yet unsurprisingly, iSIDG performs the worst among all of the methods in almost all of the datasets. On one hand, the iterative process surely has a negative effect on the time efficiency, on the other hand, iterative process encourages our method to learn the adjacency matrix more accurate than baseline methods on most of the datasets. It is notable that ACD has comparable training time to iSIDG, and has comparable accuracy with iSIDG on Kuramoto dataset.

We summarized the memory efficiency of iSIDG and baseline models in Table 16. As shown in the table, the number of total parameters of iSIDG is 0.6 M larger than NRI and fNRI, but is less than half of the number of ACD, which indicates the moderate memory efficiency of iSIDG among baseline methods.

# G  Broader Impact

Methods such as iSIDG for structural inference of dynamical systems allow for numerous researchers in the field of physics, chemistry and biology to study the interactions inside the systems. We have shown that iSIDG works well on either one-dimensional node features or multi-dimensional features, where the features are continuous variables, which proves its wide application. While the emergence of the structural inference technology may be extremely helpful for many, it has the potential for misuse. Potentially, iSIDG can be extended to infer the online social connections via measuring mutual information, which could erode privacy.