# OpenReview forum: "Iterative Structural Inference of Directed Graphs"
_NeurIPS.cc/2022/Conference — NeurIPS 2022 Accept_

### Official Review · Reviewer_cY4q · 2022-07-02

**Rating:** 6
**Confidence:** 3
**Soundness:** 3 good
**Presentation:** 3 good
**Contribution:** 3 good

**Summary:**

The paper studies the structural inference problem of reconstructing the asymmetric adjacency matrix of the graph in an unsupervised fashion. Specifically, the paper introduces an iterative process based on Variational Information Bottleneck that uses GNN-based encoder to infer directed interaction represented in the latent space, which is then used to update the asymmetric adjacency iteratively starting from a fully connected graph. The updated adjacency is decoded to predict the future states of the nodes. The paper also introduces additional regularization terms in the objective function for realistic structure and eliminate the influence of indirect interactions.

**Questions:**

- In Table 3, we can observe that the hyperparameters vary with different scales according to the dataset. Is the performance sensitive with respect to the hyperparameter choice? Why is the scale of the hyperparameter different?

- I would like to see statistical tests on the results, as some datasets (e.g. TF, Springs, Netsim 1 and 2) show large variance.

**Limitations:**

- The authors present scalability limitations in section E of the Appendix, which is a common limitation among the baselines.

**Strengths And Weaknesses:**

**Strength**
- The paper is well-written and the presentation is clear.
The preliminaries section provides a detailed explanation of the problem which the paper is trying to solve, and Figure 1 gives a clear overview of the model architecture.

- The proposed iterative scheme for structural inference seems to be novel and simple, yet effective. Although I have some concerns about the efficiency (described in the Weakness part), the proposed method shows clear performance improvement.

- The paper clearly addresses the problem of "the influence of the indirect connections" of the structural inference methods, and provides the solutions for eliminating such challenges.

- Extensive analyses are provided in the Appendix which gives further insights. Especially, the analysis of the scalability experiments shows meaningful results, even with the limitations.

**Weakness**

- The proposed method seems to be quite slow due to the iterative scheme, thus comparing time/memory efficiency may address my concern. Especially, as iSIDG shows marginal improvements on some datasets compared with fNRI (e.g. TF, Springs, Kuramoto), I believe the efficiency comparison is critical. Furthermore, the efficiency problem may be increased for larger datasets. I would like to see the efficiency comparison on Springs 100 and ESC datasets (as used in section E of Appendix).

- The influence of the hyperparameters of Equation 21 is not sufficiently explained. Although the ablation study shows the results of each term in the loss, without the analysis, the significance of the proposed hybrid loss is not fully clarified.

---

> ### Author Response · Authors · 2022-08-02
> **Response to Reviewer cY4q (Part 1)**
>
> We want to thank the reviewer for the motivating review. We are happy the reviewer found our method to be novel and simple, yet effective.
>
> Here are answers to concerns raised by the reviewer:
>
> > The proposed method seems to be quite slow due to the iterative scheme, thus comparing time/memory efficiency may address my concern. Especially, as iSIDG shows marginal improvements on some datasets compared with fNRI (e.g. TF, Springs, Kuramoto), I believe the efficiency comparison is critical. Furthermore, the efficiency problem may be increased for larger datasets. I would like to see the efficiency comparison on Springs 100 and ESC datasets (as used in section E of Appendix).
>
> Many thanks for the comment. We firstly summarized the training time of iSIDG and baseline methods on "Synthetic networks", "NetSim datasets" and "Physical simulations" in the following table: (hour)
>
> | Methods | LI   | LL   | CY   | BF   | TF   | BF-CV | NetSim1 | NetSim2 | NetSim3 | Springs | Particles | Kuramoto |
> | ------- | ---- | ---- | ---- | ---- | ---- | ----- | ------- | ------- | ------- | ------- | --------- | -------- |
> | iSIDG   | 48.2 | 50.6 | 40.8 | 44.7 | 40.3 | 44.0  | 20.7    | 36.9    | 50.8    | 42.2    | 36.0      | 39.2     |
> | NRI     | 14.3 | 18.2 | 13.0 | 15.5 | 13.6 | 16.9  | 8.8     | 16.0    | 21.5    | 20.1    | 20.3      | 19.8     |
> | fNRI    | 15.5 | 21.9 | 14.9 | 18.6 | 13.7 | 18.0  | 9.0     | 17.8    | 25.6    | 22.8    | 20.7      | 19.0     |
> | MPRI    | 5.0  | 14.4 | 3.6  | 8.0  | 5.5  | 7.9   | 2.1     | 5.6     | 9.5     | 7.9     | 6.6       | 6.3      |
> | ACD     | 40.5 | 42.8 | 39.6 | 44.0 | 41.7 | 43.2  | 20.5    | 35.8    | 45.7    | 39.8    | 36.4      | 38.0     |
>
> We then summarized the training time of iSIDG and baseline methods on "Springs100" and "ESC" datasets in the following table: (hour)
>
> | Methods | Springs100 | ESC  |
> | ------- | ---------- | ---- |
> | iSIDG   | 106.5      | 96.8 |
> | NRI     | 40.6       | 39.4 |
> | fNRI    | 49.0       | 42.0 |
> | MPRI    | 20.7       | 19.5 |
> | ACD     | 82.4       | 80.4 |
>
> All of the results are the averaged training time of 10 rounds of each method, and is summarized in hours. Unfortunately yet unsurprisingly, iSIDG performs the worst among all of the methods in almost all of the datasets. On one hand, the iterative process surely has a negative effect on the time efficiency, on the other hand, iterative process encourages our method to learn the adjacency matrix more accurate than baseline methods on most of the datasets. It is notable that ACD has comparable training time to iSIDG, and has comparable accuracy with iSIDG on Kuramoto dataset.
>
> We summarized the memory efficiency of iSIDG and baseline models in the following table:
>
> | Methods | # of Total Parameters |
> | ------- | --------------------- |
> | iSIDG   | 1.72 M                |
> | NRI     | 1.12 M                |
> | fNRI    | 1.12 M                |
> | MPRI    | 0.62 M                |
> | ACD     | 3.70 M                |
>
> where "M" represents million. As shown in the table, the number of total parameters of iSIDG is 0.6 M larger than NRI and fNRI, but is less than half of the number of ACD, which indicates the moderate memory efficiency of iSIDG among baseline methods.
>
> We added the all of the above mentioned three tables to the Section G in the appendix of revised submission.

---

> > ### Author Response · Authors · 2022-08-02
> > **Part 2**
> >
> > > The influence of the hyperparameters of Equation 21 is not sufficiently explained. Although the ablation study shows the results of each term in the loss, without the analysis, the significance of the proposed hybrid loss is not fully clarified.
> >
> > Many thanks for the comment. Due to page limitation, we haven't discussed about the hyperparameters and the terms in the hybrid loss. We would like to clarify that all of the regularization terms in the hybrid loss are leveraged to eliminate indirect connections in the learned adjacency matrix.
> >
> > Recall the Equation 21 is: $\mathcal{L} = \mathcal{L}_p + \mu \cdot\mathcal{L}_K + \alpha \cdot \Omega(\mathbf{Z},V) + \beta \cdot \mathcal{L}_d + \gamma \cdot \mathcal{L}_s$. (We denote $\mathcal{L}_K$ as the terms of KL-divergence in the original equation. The subscript with multiple letters seemed to be bugged at the moment.)
> >
> > - $\mathcal{L}_p$ and $\mathcal{L}_K$ are common terms in ELBO for VAE, where the former is used to calculate the prediction error between the output of the encoder and the target, and the latter is used to regularize the pattern in latent space. We used $\mu$ to match the two terms to the same scale.
> > - We adopt the assumption of graph signals that values change smoothly across adjacent nodes. So the Dirichlet energy $\Omega(\mathbf{A},\mathbf{X})$ (Equation 10) maps the graph representation $\mathbf{G} = (\mathbf{A}, \mathbf{X})$ to a line, so that the connected points stay as close together as possible [3]. We used $\alpha$ to match the $\Omega$ to the same scale of $\mathcal{L}_p$.
> > - The connectiveness term $\mathcal{L}_d$ has the purpose to ensure that each node has at least one edge with another node. (The result of $\mathbf{Z}\mathbf{1}$ is the node degree vector.) We use the logarithmic barrier to force the degrees to be positive, but not prevent edges from becoming zero. We used $\beta$ to match the $\mathcal{L}_d$ to the same scale of $\mathcal{L}_p$.
> > - However, adding the logarithmic term in $\mathcal{L}_d$ leads to very sparse graphs, and changing its weight in the loss ($\beta$) only changes the scale of the solution and not the sparsity pattern. For this reason, we added the third term $\mathcal{L}_s$. Yet it was mentioned in [18] and observed by us that adding an $\ell$-1 norm to control sparsity was not very useful. So we chose the Frobenius norm, which penalized the big values but not the smaller ones. This leads to a more dense adjacency matrix for a bigger value of $\gamma$.
> >
> > We also showed the count of indirect connections with different path lengths in Figure 4. During experiments, we firstly scaled the various terms in the hybrid loss to the same scale. Then we found the different values of the weights of the regularization terms in hybrid loss had a minor effect, and the number of indirect connections in the learned adjacency matrices had only a minor difference. Therefore, we only reported the learned adjacency matrices and the counts of indirect connections by setting the weights to 0.
> >
> > > In Table 3, we can observe that the hyperparameters vary with different scales according to the dataset. Is the performance sensitive with respect to the hyperparameter choice? Why is the scale of the hyperparameter different?
> >
> > Many thanks for the question!
> >
> > Yes, $\delta$ has different scales according to the dataset. As shown in Equation 16 (on page 6) and Algorithm 1 (on page 15), $\delta$ is utilized to control the stopping condition. We observed that experiments on datasets of undirected graphs converged faster than on datasets of directed graphs. So we set the value of $\delta$ on these datasets to be one scale larger.
> >
> > We leveraged $\sigma$ be the variance terms of $\mathcal{L}_p$, and the value of it differed a bit between datasets of undirected graphs and directed ones.
> >
> > $\eta$ was leveraged to control the rounds of training before the first iterative process. We reported the values as the ones to produce the best experimental results from a search space of $80$ to $200$.
> >
> > The remaining hyperparameters are the ones in hybrid loss. We set the value of which to match the corresponding terms in the loss function to the same scale.
> >
> > We tested the sensitivity of iSIDG against the choice of hyperparameters, and only the choice of $\eta$ mattered a little bit. (This issue can be solved by performing a search on it.) The rest hyperparameters had no significant effect on the performance.

---

> > > ### Author Response · Authors · 2022-08-02
> > > **Part 3**
> > >
> > > > I would like to see statistical tests on the results, as some datasets (e.g. TF, Springs, Netsim 1 and 2) show large variance.
> > >
> > > Many thanks for the comment.
> > >
> > > We performed randomized permutation tests between the results of iSIDG and other methods.
> > >
> > > For short, we describe the steps to perform a randomized permutation test:
> > >
> > > 1. Calculate AUROC of A vs. B.
> > > 2. Create $C_1$, such that $C_1$ is a pair-wisely randomly shuffled list of scores from A and B. In other words, $C_1$ is a simulation of what a random non-systematic difference looks like.
> > > 3. Measure AUROC of $C_1$.
> > > 4. Test if AUROC of $C_1$ is better than AUROC of A. If yes, increment counter $damn$.
> > > 5. Repeat step 2 to 4 nn many times, but instead of $C_1$, use $C_i$ where $i ∈\{2,3,…,n\}$.
> > > 6. Then, $p=\frac{damn}{n}$.
> > > 7. If $p≤α$, then the difference is significant. Usually $α=0.05$.
> > >
> > > We implemented it with scipy.stats.permutation_test, and report the $p$ values in the following table:
> > >
> > > | Compare   | LI     | LL     | CY     | BF     | TF     | BF-CV  | NetSim1 | NetSim2 | NetSim3 | Springs | Particles | Kuramoto |
> > > | --------- | ------ | ------ | ------ | ------ | ------ | ------ | ------- | ------- | ------- | ------- | --------- | -------- |
> > > | with NRI  | 0.0002 | 0.0002 | 0.0002 | 0.0002 | 0.0056 | 0.0002 | 0.0004  | 0.045   | 0.0002  | 0.0038  | 0.0004    | 0.0008   |
> > > | with fNRI | 0.0002 | 0.0002 | 0.0002 | 0.0006 | 0.0222 | 0.0002 | 0.0002  | 0.0002  | 0.0002  | 0.0052  | 0.0008    | 0.0056   |
> > > | with MPRI | 0.0002 | 0.0002 | 0.0002 | 0.0002 | 0.0002 | 0.0002 | 0.0002  | 0.0002  | 0.0002  | 0.002   | 0.0002    | 0.0002   |
> > > | with ACD  | 0.0002 | 0.0004 | 0.0002 | 0.0002 | 0.0006 | 0.0002 | 0.0002  | 0.0002  | 0.0004  | 0.0132  | 0.0006    | 0.0954   |
> > >
> > > In the table, the smaller the $p$ value is, the larger the difference between two datasets is. And the results of the randomized permutation test match the reported results in our paper.
> > >
> > > ------------------
> > > ### References:
> > >
> > > [3] M. Belkin and P. Niyogi. Laplacian eigenmaps and spectral techniques for embedding and clustering. In Proceedings of the 15th International Conference on Neural Information Processing Systems (NIPS), pages 585–591, 2001.
> > >
> > > [18] V. Kalofolias. How to learn a graph from smooth signals. In Artificial Intelligence and Statistics, pages 920–929. PMLR, 2016.

---

### Official Review · Reviewer_yhZf · 2022-07-04

**Rating:** 5
**Confidence:** 3
**Soundness:** 3 good
**Presentation:** 3 good
**Contribution:** 2 fair

**Summary:**

The authors propose the Iterative Structural Inference of Directed Graphs (iSIDG), which is able to infer the directed interactions between agents in dynamic systems with information bottleneck theory and extra regularization. The proposed framework is examined on several simulated and synthetic networks to identify the interactions.

**Questions:**

- In the problem formulation, the network structure (adjacency matrix) is considered as consistent across time, while only the node feature is changing. In many real-world datasets, the pairwise connection can appear and disappear over the time and new nodes can also join at some time point. Can the author further explain how their framework can be applied to such scenarios?
- What is the model complexity here? I would like to see the duration of model training compared with baselines. It also helps to see if the proposed method can be scaled up to large complex systems.


**Limitations:**

See Weakness

**Strengths And Weaknesses:**

Strengths:
- This paper focuses on inferring the underlying structure of agent interactions in time-varying systems, which is a complex problem due to the dynamic feature.
- Apart from using information bottleneck of VAE to infer the structure, the proposed method also consider the sparsity and connectiveness of the asymmetric adjacency matrix property and design extra regularization term to address the related features. Ablation study also shows that the sparsity regularization term can especially help to infer direct connections.

Weakness:
* Size of the dataset (number of nodes) is very small here. Is it because of the limitation of the datasets or the proposed framework is very computationally expensive?
* It would be beneficial if the authors add a ground truth subfigure in Fig. 2.
* The proposed method is only applied on simulated or synthetic datasets. I strongly recommend examining the framework on at least two real-world datasets to show the priority of the proposed method to reconstruct the graph structure. Simulated or synthetic data are usually more "predictable" since the data generation process is known and with less uncertainty, while high complexity and unknowns is more likely to be involved in real-world datasets.
* I suggest the authors to look into the following related paper on reconstructing network
* * https://openreview.net/forum?id=rJgTciR9tm
* * https://www.nature.com/articles/s41467-019-09774-x

---

> ### Author Response · Authors · 2022-08-02
> **Response to Reviewer yhZf (Part 1)**
>
> First of all, we want to thank the reviewer for the thoughtful and helpful review. We are happy the reviewer found iSIDG to be an important research problem and iSIDG to be effective to infer direct connections.
>
>
> Here are answers to concerns raised by the reviewer:
>
> > Size of the dataset (number of nodes) is very small here. Is it because of the limitation of the datasets or the proposed framework is very computationally expensive?
>
> Yes and yes.
>
> The datasets we mentioned in the main body of our submission were either benchmark in [33], or used in previous works [22, 30, 51]. These datasets are widely used in the research field of structural inference. Our idea was to design an algorithm with higher accuracy than previous methods and thus we have tested iSIDG on these datasets for comparison. Besides that, we also tested iSIDG on large networks (with 96 and 100 nodes), and summarized the results in Table 10 in the appendix. Our iSIDG outperforms baseline methods even more on these two large datasets. We agree that scalability is an issue of iSIDG. Yet, it is also a common issue of all baselines with comparable inference accuracy. Scalability of structural inference methods is widely recognized as a research challenge in the research community (also acknowledged by reviewer 3).
>
> One main limitation of iSIDG is the requirement of a fully connected initial adjacency matrix, which makes the algorithm (especially the encoder) to be computationally expensive. As mentioned in Section B.1 of the appendix, we used a single NVIDIA Tesla V100 SXM2 graphic card to conduct experiments, and had to decrease the batch size for the datasets with more than 10 nodes. How to improve the scalability of iSIDG is our next step in research.
>
> > It would be beneficial if the authors add a ground truth subfigure in Fig. 2.
>
> Many thanks for the advice. Because of the page limitation, we have only put the ground truth of "LI" in the subfigure of Figure 5 in the appendix. We revised the submission by stating the ground truth of "LI" can be found in Figure 5. If desired, we will try to fit the ground truth subfigure in the main body of the camera-ready version.
>
> > The proposed method is only applied on simulated or synthetic datasets. I strongly recommend examining the framework on at least two real-world datasets to show the priority of the proposed method to reconstruct the graph structure. Simulated or synthetic data are usually more "predictable" since the data generation process is known and with less uncertainty, while high complexity and unknowns is more likely to be involved in real-world datasets.
>
> Many thanks for the thoughtful advice. As we stated in the answer to the first question, because the "Synthetic Networks", "NetSims Datasets" and "Physical Simulations" are widely used in several works to test the accuracy of structural inference [22, 30, 33, 51], we tested iSIDG on these datasets to show its effectiveness (i.e., high inference accuracy). Besides that, we actually have conducted experiments of iSIDG on a real-world dataset of profiled embryonic stem cells (ESC) [4] in Section E of the appendix. ESC is generated using quantitative PCR on real cells, and has 96 nodes and 409 edges. We summarized the experimental results of iSIDG and baseline methods in Section E and Table 10.
>
> Moreover, we downloaded the dataset of hematopoietic stem cells (HSC) [31], which was also generated using quantitative PCR from 48.48 array chips. HSC has 33 nodes and 126 edges. We ran extra experiments on the HSC dataset and we present the averaged AUROC results in % of iSIDG and baselines in the following table:
>
> | Method | HSC  |
> | ------ | ---- |
> | iSIDG  | 58.6 |
> | NRI    | 42.9 |
> | fNRI   | 45.0 |
> | MPIR   | 37.6 |
> | ACD    | 44.3 |
>
> As shown in the table above and Table 10, iSIDG outperforms other baseline methods on the two real-world datasets: ESC and HSC. We believe these experiments demonstrate the effectiveness of iSIDG on real-world data. Due to the page limitation, we will move section E to the main body in the camera-ready version.
>
> >I suggest the authors to look into the following related paper on reconstructing network
> >
> >- https://openreview.net/forum?id=rJgTciR9tm
> >- https://www.nature.com/articles/s41467-019-09774-x
>
> Many thanks for the suggestion! After reading the listed papers, we found these papers to be full of inspiration. We are especially interested in how to combine causality with the research problem of structural inference, which is stated in the second paper of the list. However, the problem setting of the second paper is still different from ours, since iSIDG has to infer the entire adjacency matrix without any knowledge about interactions. We revised the section on related works to discuss this paper.

---

> > ### Author Response · Authors · 2022-08-02
> > **Part 2**
> >
> > > In the problem formulation, the network structure (adjacency matrix) is considered as consistent across time, while only the node feature is changing. In many real-world datasets, the pairwise connection can appear and disappear over the time and new nodes can also join at some time point. Can the author further explain how their framework can be applied to such scenarios?
> >
> > Thank you for this excellent question! The difficulty of applying iSIDG to the scenarios mentioned above consists of two key points: (1) the fixed latent space and (2) the iterative process. In the following, we discuss two ways on how to adapt iSIDG to address these issues:
> >
> > 1. We can adopt the DVAE methods mentioned in https://arxiv.org/pdf/2008.12595.pdf, which leverages a temporal chain or a causal model to capture the changes, and the latent space of which changes from frame to frame. However, we believe that it is useless to feed a fixed learned adjacency matrix in the iterative process, since the adjacency matrix keeps changing over time. We would like the model to learn a "global state" on the adjacency matrix, and feed the state back during training. (We can imagine the "global state" as the global attributes mentioned in https://arxiv.org/abs/1806.01261).
> >
> > 2. We can build a large initial adjacency matrix, which also reserves a mark for the nodes joining later on. We then build another indication matrix to indicate which nodes appear in the current frame. Suppose we have a random length of frames with mark $[t_a, t_b]$, where $t_a$ and $t_b$ mark the beginning of the change (either changes in edges or the join of new nodes), and the end of the change, respectively. In a single process of iterative training, we feed the data with the segmentations of data $\{X_t, t \in [t_a, t_b]\}$ and the corresponding indication matrix to train the model frame-to-frame. We feed the averaged adjacency matrix back to the input of the encoder in the next iteration. (We may have to design a weighted sum of all adjacency matrices from the frames, in order to preserve the dynamical changes of edges fairly.)
> >
> > Note that it is necessary for both of the methods that the data should capture the frame when edges appear/disappear or new nodes join.
> >
> > We believe that designing a structural inference framework for dynamical graphs is also of great importance, and would like to present it as our future work.
> >
> > >What is the model complexity here? I would like to see the duration of model training compared with baselines. It also helps to see if the proposed method can be scaled up to large complex systems.
> >
> > We summarize the duration of model training (in hours) in the following table (averaged results of 10 rounds):
> >
> > | Methods | LI   | LL   | CY   | BF   | TF   | BF-CV | NetSim1 | NetSim2 | NetSim3 | Springs | Particles | Kuramoto |
> > | ------- | ---- | ---- | ---- | ---- | ---- | ----- | ------- | ------- | ------- | ------- | --------- | -------- |
> > | iSIDG   | 48.2 | 50.6 | 40.8 | 44.7 | 40.3 | 44.0  | 20.7    | 36.9    | 50.8    | 42.2    | 36.0      | 39.2     |
> > | NRI     | 14.3 | 18.2 | 13.0 | 15.5 | 13.6 | 16.9  | 8.8     | 16.0    | 21.5    | 20.1    | 20.3      | 19.8     |
> > | fNRI    | 15.5 | 21.9 | 14.9 | 18.6 | 13.7 | 18.0  | 9.0     | 17.8    | 25.6    | 22.8    | 20.7      | 19.0     |
> > | MPRI    | 5.0  | 14.4 | 3.6  | 8.0  | 5.5  | 7.9   | 2.1     | 5.6     | 9.5     | 7.9     | 6.6       | 6.3      |
> > | ACD     | 40.5 | 42.8 | 39.6 | 44.0 | 41.7 | 43.2  | 20.5    | 35.8    | 45.7    | 39.8    | 36.4      | 38.0     |
> >
> > As for larger datasets, we tested "Springs100", "ESC", and the "HSC" mentioned in the response letter. We summarize the duration of model training (in hours) in the following table (averaged results of 10 rounds for "Springs100" and "ESC"; averaged results of 3 rounds for "HSC").
> >
> > | Methods | Springs100 | ESC  | HSC  |
> > | ------- | ---------- | ---- | ---- |
> > | iSIDG   | 106.5      | 96.8 | 50.3 |
> > | NRI     | 40.6       | 39.4 | 30.4 |
> > | fNRI    | 49.0       | 42.0 | 31.8 |
> > | MPRI    | 20.7       | 19.5 | 12.0 |
> > | ACD     | 82.4       | 80.4 | 51.8 |
> >
> > According to the results presented above, iSIDG seems to suffer from the problem of scalability, which is also the case for other VAE-based methods (NRI, fNRI and ACD). We mentioned this as the limitation of iSIDG in the conclusion section of our paper. The iterative process of iSIDG leads to more time consumption during model training (still comparable with ACD). Our main goal of this work is to design an **effective** method for structural inference: iSIDG achieves more accurate inference results than baseline methods on most of the datasets; and for larger networks the inference accuracy of iSIDG gets even better than any of the baseline methods.
> >
> > We added the run-time summary to Section G in the appendix of our paper.

---

> > > ### Author Response · Authors · 2022-08-02
> > > **References**
> > >
> > > [3] M. Belkin and P. Niyogi. Laplacian eigenmaps and spectral techniques for embedding and clustering. In Proceedings of the 15th International Conference on Neural Information Processing Systems (NIPS), pages 585–591, 2001.
> > >
> > > [22] T. Kipf, E. Fetaya, K.-C. Wang, M. Welling, and R. Zemel. Neural relational inference for interacting systems. In Proceedings of the 35th International Conference on Machine Learning (ICML), pages 2688–2697. PMLR, 2018.
> > >
> > > [30] S. Löwe, D. Madras, R. Zemel, and M. Welling. Amortized causal discovery: Learning to infer causal graphs from time-series data. arXiv preprint arXiv:2006.10833, 2020.
> > >
> > > [33] A. Pratapa, A. P. Jalihal, J. N. Law, A. Bharadwaj, and T. Murali. Benchmarking algorithms for gene regulatory network inference from single-cell transcriptomic data. Nature Methods, 17(2): 147–154, 2020.
> > >
> > > [51] E. Webb, B. Day, H. Andres-Terre, and P. Lió. Factorised neural relational inference for multi-interaction systems. arXiv preprints arXiv:1905.08721, 2019.

---

> > > ### Comment · Reviewer_yhZf · 2022-08-09
> > > **Thanks for your response**
> > >
> > > Thanks for the authors' response. My major questions about the dataset size etc. have been answered. It is understandable given the fact that the dataset are benchmark datasets or from previous works. I would like to raise my rating score, but I still recommend the authors to look into the variational problem formulation with non-fully connected graphs. This will allow more freedom to fit into the real-world scenerio where the graph is larger and more heterogeneous.

---

> > > > ### Author Response · Authors · 2022-08-09
> > > > **Thanks!**
> > > >
> > > > Thanks for your response! We couldn't agree more: the variational problem formulation with non-fully connected graphs you suggested is a great idea, since it allows the structural inference algorithm more freedom for real-world scenarios. We will revise the paper and take this precise comment into account.

---

### Official Review · Reviewer_3CVG · 2022-07-09

**Rating:** 6
**Confidence:** 2
**Soundness:** 3 good
**Presentation:** 3 good
**Contribution:** 2 fair

**Summary:**

This paper proposed a variational model to infer the underlying directed graph of a dynamical system from observed features of the agents over a time period. Regularization terms were also introduced to encourage learing of directed interactions. Evaluations on a benchmark of datasets show good results.

**Questions:**

1. How sensitive is this approach to the initialization of the adjacency matrix A? Why the fully connected one is a good choice?
2. What kind of GNN was used in your experiments? Does the choice of GNN matter?
3. Is your hybrid loss strategy applicable to other baseline approach?

**Limitations:**

Yes they do.

**Strengths And Weaknesses:**

Strengths:

1. The idea of iterative learning of the adjacency matrix seems new.


Weaknesses:

1. There are many regularization terms in this approach, and all of them seems to be crucial for the overall performance of direct interation discovery, which makes the information bottleneck formulation a little bit vague.

2. It is not clear how the inner loop is conducted in section 4.4.

---

> ### Author Response · Authors · 2022-08-02
> **Response to Reviewer 3CVG (Part 1)**
>
> We would like to thank the reviewer for the thoughtful feedback. We are happy the reviewer found our idea of iterative learning of the adjacency matrix novel.
>
>
>
> Here are answers to concerns raised by the reviewer:
>
> > There are many regularization terms in this approach, and all of them seem to be crucial for the overall performance of direct interaction discovery, which makes the information bottleneck formulation a little bit vague.
>
> As stated in Section A.2 in the appendix, iSIDG performs structural inference with three mechanisms: (1) VAE, (2) iterative process, and (3) regularization terms. Among these mechanisms, VAE and iterative processes are closely bound with the formulation of the information bottleneck.
>
> - VAE leverages information bottleneck to extract the minimum sufficient statistic (the candidate of the adjacency matrix) by correctly identifying non-connections.
> - The iterative process creates a tighter bound (Equation 22) than Equation 3, which encourages iSIDG to focus on distinguishing between indirect connections and direct connections.
> - The regularization terms are utilized to incentivize and accelerate the process of eliminating indirect connections in the learned adjacency matrix.
>
> So by design, iSIDG **primarily** leverages the information bottleneck theory to eliminate the surely non-connections in the learned adjacency matrix, as well as to partially eliminate indirect connections with the help of iterative process. Then regularization terms encourage iSIDG to learn a tighter representation of the adjacency matrix in its latent space, in order to **further** eliminate the indirect connections from it.
>
>
>
> > It is not clear how the inner loop is conducted in section 4.4.
>
> We are sorry that we made the statement misleading. The "inner loop" or "inner iteration" refers to another iteration with the updated adjacency matrices. We revised the paragraphs above and below Equation 16 for clearer exposition.
>
>
>
> > How sensitive is this approach to the initialization of the adjacency matrix A? Why the fully connected one is a good choice?
>
> Thank you for this excellent question! We ran extra experiments with different initialization of the adjacency matrix A, where the probability of the existence of each edge was sampled from a Bernoulli distribution: $P(a_{ij}) \sim \text{Ber}(p)$, where $p \in$ {$0.1, 0.3, 0.5, 0.7, 0.9$} (a set of variables). We ran the experiments on "LI" and "Springs" datasets five times with the same $p$ value, while maintaining the rest settings as the same of iSIDG. The results at 2000 epochs (averaged AUROC in %) are summarized below:
>
>
>
> | p       | 0.1  | 0.3  | 0.5  | 0.7  | 0.9  | 1.0 (Fully-connected) |
> | ------- | ---- | ---- | ---- | ---- | ---- | --------------------- |
> | LI      | 80.5 | 82.8 | 82.9 | 84.3 | 86.0 | 86.2                  |
> | Springs | 85.9 | 87.8 | 88.0 | 88.6 | 89.3 | 90.5                  |
>
> We observed that the initialization of the adjacency matrix A has an effect on the final results. We continued running the training sessions, and all of the models with different $p$ values performed equally as iSIDG after running extra 200 - 450 epochs. We can reach the conclusion that the convergence time of the training process is positively associated with the proportion of "1"s in the initial adjacency matrix. A fully-connected initial adjacency matrix leads to the fastest convergence.
>
>
> > What kind of GNN was used in your experiments? Does the choice of GNN matter?
>
> We are sorry for not mentioning the choice of GNN in the main body of our submission. We used GIN in the encoder. We added one sentence to state the choice of GNN in the encoder in Section 4.2 in our revision.
>
> Yes. We have conducted experiments on the choice of GNN among GCN, GIN, and GAT in Section D.4 (in the appendix, because of page limitation), and the results are shown in Table 9. From Table 9, we can see that GIN outperforms other alternatives.

---

> > ### Author Response · Authors · 2022-08-02
> > **Part 2**
> >
> > > Is your hybrid loss strategy applicable to other baseline approach?
> >
> > Yes, our hybrid loss strategy is theoretically applicable to NRI, fNRI, and ACD, which use a VAE as the backbone. For MPIR, unfortunately, we cannot apply the hybrid loss to it. We actually conducted experiments to test the effectiveness of the hybrid loss on NRI, fNRI, and ACD, and observed that the AUROC results of NRI and ACD increased about 2 - 3 % on average, while we could not observe a notable increase in the results of fNRI. The reason is that fNRI has to learn the topology of non-edges and edges in two separate latent spaces, and the assignment of each latent space to its edge-type (whether it represents the edges or non-edges) requires ground truth during test and inference. As we had no idea about which latent space was for the real adjacency matrix during training, we applied the hybrid loss on both latent spaces, and caused a negative effect on the one supposed to represent the non-edges by encouraging sparser formatting.
> >
> > If desired, we will add the results of the NRI, fNRI, and ACD with hybrid losses in the appendix.

---

### Author Response · Authors · 2022-08-09
**Author summary of rebuttal discussion**

We would like to thank all three reviewers for their time and valuable feedback!



We appreciate their assessment of our work as a **"novel and simple, yet effective"**, **"well-written and the presentation is clear"** (cY4q), **"the idea of iterative learning of the adjacency matrix is new"** (3CVG), the focus of the work is **"a complex problem due to the dynamic feature"** and **"the sparsity regularization terms can especially help to infer direct connections"** (yhZf). We are also motivated that the reviewers have the opinion that our paper **"clearly addresses the problem of 'the influence of the indirect connections' of the structural inference methods"**, and addressing **"extensive analyses are provided in the Appendix which gives further insights"** (cY4q).



We received numerous helpful suggestions; this is a summary of the main concerns and how we addressed them:

- In response to reviewer 3CVG, we designed and ran experiments where the adjacency matrix was initialized differently, and showed that the initialization of the adjacency matrix had an impact on the convergence time.

- In response to reviewer yhZf, we firstly clarified that because "Synthetic Networks", "NetSims Datasets" and "Physical Simulations" are widely used in several works, we ran iSIDG on these datasets to show its effectiveness (i.e., high inference accuracy). We then downloaded the dataset of hematopoietic stem cells (HSC) and ran iSIDG on the dataset. Together with the experimental results in our appendix on ESC, the results on HSC demonstrated the effectiveness of iSIDG on real-world data.

- Interestingly, in response to reviewer yhZf, we shortly described our idea on how to adapt iSIDG to dynamic graphs.

- In response to reviewers yhZf and cY4q, we added a section in the appendix on time/memory efficiency, which showed the computational time and model sizes of iSIDG and baseline methods.

- In response to reviewer cY4q, we briefly explained the importance of the hyperparameters and each term in the hybrid loss. Besides that, we also performed randomized permutation tests on the AUROC results of iSIDG and baseline methods, which further supported the performance of iSIDG in terms of inference accuracy.



All reviewers have already taken the time to consider our responses and largely indicated being happy with them. We are thankful for reviewer yhZf on the idea of how to perform structural inference on real-world scenarios and heterogeneous graphs.

We are looking forward to receiving further comments on our response and suggestions on our work. In case we cannot address the new comments and suggestions during the discussion period, we would like to address them in camera-ready version of our work.

Thanks again to all reviewers for their time, effort, and constructive feedback!

---

### Meta-Review · Area_Chair_SiLa · 2022-08-29

**Recommendation:** Accept
**Confidence:** Less certain

**Metareview:**

This is a bordeline paper.

All reviewers liked the paper but were a bit concerned with the number of regularization terms (the loss combines VAE and information bottleneck approaches) and thus the number of hyperparameters. Secondly, the reviewers were also initially concerned with the size of the benchmarks but the discussion phase convinced them that these are standard in the field. Thirdly, some reviewers pointed to more related work. This, however, was not a crucial point.

Since all reviewers finds the paper interesting and well-written, acceptance is recommended.

**Award:**

No

---

### Decision · Program_Chairs · 2022-09-14

Accept